# Dual-wavelength switchable single-mode lasing from a lanthanide-doped resonator

Limin Jin [1,5✉], Xian Chen [2,5✉], Yunkai Wu [1,5], Xiangzhe Ai [2], Xiaoli Yang [1], Shumin Xiao [1,3,4✉] & Qinghai Song [1,3,4✉]

The development of multi-wavelength lasing, particularly with the wavelength tuning in a wide spectral range, is challenging but highly desirable for integrated photonic devices due to its dynamic switching functionality, high spectral purity and contrast. Here, we propose a general strategy, that relies on the simultaneous design on the electronic states and the optical states, to demonstrate dynamically switchable single-mode lasing spanning beyond the record range (300 nm). This is achieved through integrating the reversely designed nanocrystals with two size-mismatched coupled microcavities. We show an experimental validation of a crosstalk-free violet-to-red single-mode behavior through collective control of asymmetric excitation and excitation wavelength. The single-mode action persists for a wide power range, and presents significant enhancement when compared with that in the microdisk laser. These findings enlighten the reverse design of luminescent materials. Given the remarkable doping flexibility, our results may create new opportunities in a variety of frontier applications.

[1] Ministry of Industry and Information Technology Key Lab of Micro-Nano Optoelectronic Information System, Harbin Institute of Technology, Shenzhen 518055, P. R. China. [2] College of Materials Science of Engineering, Shenzhen University, Shenzhen 518060, P. R. China. [3] Collaborative Innovation Center of Extreme Optics, Shanxi University, Taiyuan 030006 Shanxi, P. R. China. [4] Pengcheng Laboratory, Shenzhen 518055, P. R. China. [5] These authors contributed equally: Limin Jin, Xian Chen, Yunkai Wu. ✉email: jinlm2011@126.com; x.chen87@outlook.com; shumin.xiao@hit.edu.cn; qinghai.song@hit.edu.cn

**M**ulti-wavelength lasing from a single luminescent material capable of emission wavelengths over a wide spectral range fulfills diverse applications such as general lighting, high-throughput sensing, multicolor imaging, laser display, high-level security, on-chip recording, and optical communication[1–5]. The ultimate form of such laser lies in a type of crosstalk-free tunable single-mode operation with high contrast and reversibility in a wide tuning range, or even across the entire violet and visible spectrum[6–8].

Such miniaturized lasers can be achieved by exploiting a wavelength-tunable gain medium, that is effectively coupled into a microcavity[6,7,9,10]. Among them, free-standing semiconductor alloy nanowires have received considerable attention, where the nanowires serve as both the gain medium and optical cavity. For example, Ning et al. have demonstrated a dual red-to-green single-mode lasing in a monolithic CdSSe alloy nanowire with the wide-gap end rolled up to a q-like shape[6]. This strategy by controlling the composition gradient of a single heterostructure was also adopted to construct CsPbCl$_{3-3x}$Br$_{3x}$ perovskite nanowires, CdS$_x$Se$_{1-x}$ nanoribbons and metal-organic framework microcrystals (MOFs)[7,9,10]. Besides, the coupled microcavities[4,5,11–14], including heterogeneously coupled organic nanowires, homogeneously coupled ZnO microrods, folded CdSe nanowires and two mutually coupled microrings have been substantiated to generate a wavelength-tunable single-mode lasing based on either the Vernier effect or the parity-time symmetry breaking mechanisms, respectively. Nonetheless, those approaches usually require precise manipulation at the nanoscale. The remote dynamic modulation method is complementary to the above approaches. For instance, the dependence of lasing wavelengths on external stimuli, involving temperature, electric field, refractive index environment, and chemical surroundings, have been reported in many luminescent material systems, containing perovskites, organic molecules, polymers and so on[4,8,15–23]. However, due to the limited availability of the active gain regions, the wavelength tuning range that can be realized from a single gain medium is quite limited in practice[4–23].

Upconversion nanocrystals (UCNCs), featured with plentiful energy levels and tailorable upconversion process, show great potential for switchable microlasers[24–28]. Particularly, such materials capable of tunable light spanning the full spectrum from ultraviolet to near-infrared (NIR)[2,29,30] are of significantly technological importance. Despite those advances, success in developing switchable Ln$^{3+}$-based chip-integrated single-mode laser is extremely limited by their multiwavelength and multimode outputs[2,31–34]. Therefore, it is vital to explore a novel mechanistic strategy that can generate broadband switchable single-mode lasing in mass-manufactural cavities.

In this work, we demonstrate that a UCNCs-doped size-mismatched photonic molecules (PMs) structure, evoking constructive Vernier effect in the resonance between the gain microcavity and loss one upon asymmetric pumping, gives rise to excellent unidirectional single-mode lasing. By harnessing Yb$^{3+}$-Nd$^{3+}$ co-doped UCNCs as the gain medium, dynamic lasing switch with an extremely large wavelength shift up to 300 nm is realized by external excitation modulation. This PMs laser in submillimeter scale was made by the CMOS-compatible photolithography technique, providing a smart and robust design to construct a mass-manufacturable compact lasing switch for environmental and photonic applications.

## Results

Figure 1 shows the working mechanism of the proposed UCNCs-based microlasers, which integrates two key advances of the design on the electronic states and the optical states to support

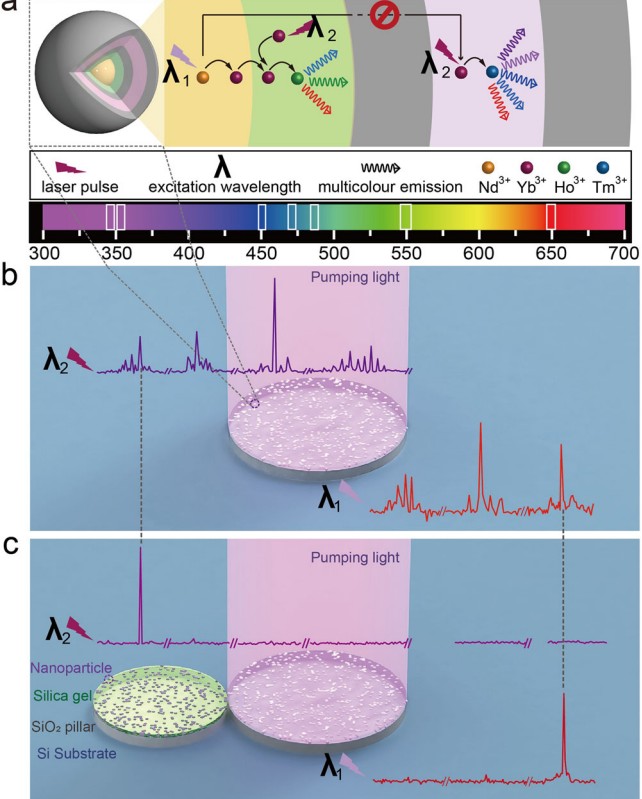

**Fig. 1 Mode-switching concept of the UCNCs-based PMs laser.**
**a** Illustration of Ln$^{3+}$-doped multi-shell nanocrystals with two distinct energy transfer paths under the excitation of λ$_1$ and λ$_2$ lasers. The distinct layers are highlighted by different colors. **b** An isolated UCNCs-based microdisk simultaneously supports multimode lasing emission with the emission wavelengths spanning over a wide spectral range. **c** UCNCs-based PMs structure is used to achieve dual-wavelength switchable single-mode laser under λ$_1$/λ$_2$ pumping across an extremely wide spectral range.

dynamically switchable single-mode lasing. The former one is accomplished through light manipulation at nanoscale (Fig. 1a). It is realized by dispatching different Ln$^{3+}$ ions into the separated layers of multi-shell UCNCs, thus achieving two distinct upconversion process under the excitation of λ$_1$ and λ$_2$ lasers, respectively. When pumping at different wavelengths, the integration of those UCNCs and microdisk resonators (Fig. 1b) from the preformed substrate[32] leads to two groups of multiwavelength lasing emission through the formation of whispering gallery modes (WGMs). This arrangement typically has an inherently isotropic output and a multiple mode feature arising from the simple microdisk laser. Generally, the single-mode operation could be achieved by decreasing the cavity size to several or a dozen of microns according to the following equation[31–34]:

$$FSR = \frac{\lambda_0^2}{\pi d n_{eff}} \qquad (1)$$

where FSR is the free spectral range, λ$_0$ is the wavelength of the resonant peak, n$_{eff}$ is the group refractive index, and d is the diameter of the microdisk resonator[2,32]. However, this strategy is quite limited since it usually gives rise to a highly increased lasing threshold and does not effectively address the multiwavelength output problem. To overcome these obstacles, the size-mismatched PMs with the cavity size at the submillimeter scale can be exploited to precisely control the optical states of UCNCs (Fig. 1c). Once the PMs lasing is established by withholding the

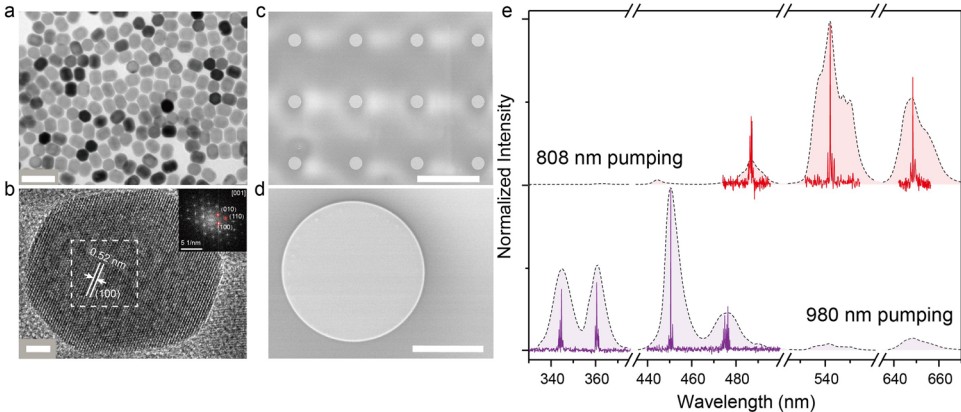

**Fig. 2 Realization of UCNCs-based microdisk lasers. a** TEM and (**b**) High-resolution TEM images of the NaGdF$_4$:Yb/Nd@NaGdF$_4$:Yb/Ho@NaYF$_4$:Ca@Na YbF$_4$:Tm@NaYF$_4$:Ca multi-shell nanocrystal. The scale bars are 100 nm and 5 nm, respectively. The inset gives the Fourier transform diffraction pattern of the region indicated in (**b**). SEM images of (**c**) a typical set of UCNCs-based microdisk array, and (**d**) an isolated one (d = ~100 μm, and t (thickness) = ~300 nm). The scale bars are 500 μm and 50 μm, respectively. **e** The observation of PL spectra (the dotted lines) from the UCNCs solution under CW 808/980 nm excitation at a power of 20 W cm$^{-2}$, respectively. The multiwavelength multimode lasing spectra (the solid red/violet lines) from the UCNCs-based microdisk laser in (**d**) pumped by 808/980 nm pulsed lasers at a power of 93 mJ cm$^{-2}$, respectively.

excitation from one of two tangent microdisks, a single-mode laser emerges from the two coupled microdisk cavities, which contrasts starkly with the microdisk lasers. Other competing modes are highly suppressed except for the lasing emission in the ultraviolet and red regions. Most importantly, with external excitation manipulation, this on-chip PMs device offers robust dynamic mode-switching functionality without the need for any other intricate components.

As a proof of concept, NaGdF$_4$:Yb/Nd(40/40)@NaGdF$_4$:Yb/ Ho(48/2)@NaYF$_4$:Ca@NaYbF$_4$: Tm(1)@NaYF$_4$:Ca core-multishell UCNCs were synthesized via a modified literature procedure[2]. In the conceptual design (Fig. S1 in Supporting Information), two groups of energy transfer paths including Nd$^{3+}$→Yb$^{3+}$→Ho$^{3+}$ and Yb$^{3+}$→Tm$^{3+}$ are incorporated into the multi-shell nanocrystals. Obviously, the Nd$^{3+}$/Yb$^{3+}$ sensitizer-pair leads to two groups of characteristic peaks of Ho$^{3+}$ and Tm$^{3+}$ ions while being optically pumped at the wavelengths of 808 nm and 980 nm, respectively. Note that the NaYF$_4$:Ca inner layer with a thickness of 5 nm is used to restrain the Nd$^{3+}$→Yb$^{3+}$→Tm$^{3+}$ energy-mediated upconversion in the 808 nm laser-excited UCNCs. The outmost layer of NaYF$_4$:Ca with a thickness of 4 nm is employed to suppress surface quenching and protect the upconversion process (Figs. S1 and S2). In addition, Nd$^{3+}$ and Ho$^{3+}$ ions are dispersed in the separated layers to avoid deleterious energy transfer between Nd$^{3+}$ and the activator ions[35], while Yb$^{3+}$ clusters are conducted in the neighboring layer to mediate the Nd$^{3+}$-sensitized upconversion process[35–37]. Thus, an efficient emission from the Ho$^{3+}$ transitions can be obtained under the excitation of 808 nm laser due to the spatially confined Nd$^{3+}$/Ho$^{3+}$ ions and the effective Nd$^{3+}$→Yb$^{3+}$ energy transfer (i.e., up to 70%, depending on the proportion of contributing Nd$^{3+}$ among all Nd$^{3+}$ ions)[35–37]. Moreover, together with the inert inner-shell, the doping amount of Yb$^{3+}$ ions in the multi-shell nanoarchitecture is optimized to minimize the emission of the Ho$^{3+}$ transitions and enhance the emission of the Tm$^{3+}$ transitions for the 980 nm laser-excited case, therefore leading to a crosstalk-free dual-mode emission profiles (see the dotted lines in Fig. 1e) under the excitation of 808 nm and 980 nm lasers.

Accordingly, NaGdF$_4$:Yb/Nd core nanocrystals with a diameter of ~16 nm (Fig. S3 in Supporting Information) were firstly prepared followed by successive deposition of four epitaxial shells of NaLnF$_4$ (Ln=Gd, Yb, Ho, Tm, Y). The successful synthesis of core-multishell UCNCs can be inferred from the results of transmission electron microscopy (TEM, Fig. 2a and S3), X-ray

powder diffraction (Fig. S4), and photoluminescence (PL, see the dotted lines in Fig. 2e). The monodispersed UCNCs were examined to have a uniform size of ~40 nm, showing a single-crystalline nature with the Bragg diffraction lines expected for the hexagonal-form NaYF$_4$ (JCPDS #16-0334). As shown in Fig. 2b, high-resolution TEM image and the Fourier transform diffraction pattern permit resolution of lattice fringes of {100} with a $d$-spacing of 0.52 nm, which matches with that of β-NaYF$_4$[2,25,30,32]. For optical characterization, the UCNCs solution was measured under the excitation of a continue-wave (CW) NIR laser. Typically, for 808 nm pumping case, the excitation energy is harvested by Nd$^{3+}$ ions incorporated in the core layer, and then migrates to Ho$^{3+}$ ions in the first shell layer through Yb-sublattice[35,36], while Tm$^{3+}$ ions dominantly exhaust the excitation energy collected by highly doped Yb$^{3+}$ ions under the condition of 980 nm irradiation. As shown in Fig. 2e, these emission peaks can be assigned to $^5F_3 \rightarrow {}^5I_8$ (487 nm), $^5S_2 \rightarrow {}^5I_8$ (542 nm), and $^5F_5 \rightarrow {}^5I_8$ (646 nm) transition of Ho$^{3+}$ ions, and $^1I_6 \rightarrow {}^3F_4$ (346 nm), $^1D_2 \rightarrow {}^3H_6$ and $^3F_4$ (362 and 451 nm), and $^1G_4 \rightarrow {}^3H_6$ and $^3F_4$ (476 and 648 nm) transition of Tm$^{3+}$ ions, respectively. Under 980 nm pumping, the weak peaks at 542 nm and 646 nm may arise from the Yb$^{3+}$→Ho$^{3+}$ upconversion process. This unique dual-mode PL behavior of such UCNC make it a promising candidate for constructing a dynamically switchable microlaser.

Subsequently, we fabricated an UCNCs-based microdisk array by simply spin-coating a mixture of UCNCs and silica resin (6.5 wt%, see Fig. S5 in Supporting Information) onto the patterned SiO$_2$ substrate[32]. Figure 2c, d shows the scanning electron microscopy (SEM) images of the UCNCs-based microdisk array and an isolated one (d = ~100 μm, t = ~300 nm), respectively. The UCNCs-based microdisk well inherits the pattern of the underneath SiO$_2$ pillar. For its lasing characterization, a NIR pulsed laser (808/980 nm, pulse width 6 ns, repetition rate 10 Hz, Φ8 mm) is focused onto the top surface of the microdisk, with the emission light from the cavity boundary being collected by an optical fiber. Figures S6 and S7 summarize the lasing action in UCNCs-based microdisk under NIR excitation. For instance, several periodic sharp peaks centered at 345.8 nm emerge from the broad emission band above the lasing threshold (i.e., P$_{th}$ = 65.68 mJ cm$^{-2}$@345.8 nm, as reflected by the second kink value in Fig. S6b), and then quickly dominate the emission spectra with increasing power. The experimental FSR reads as ~0.24 nm, which matches the calculated one from the Eq. (1)[31–34]. The

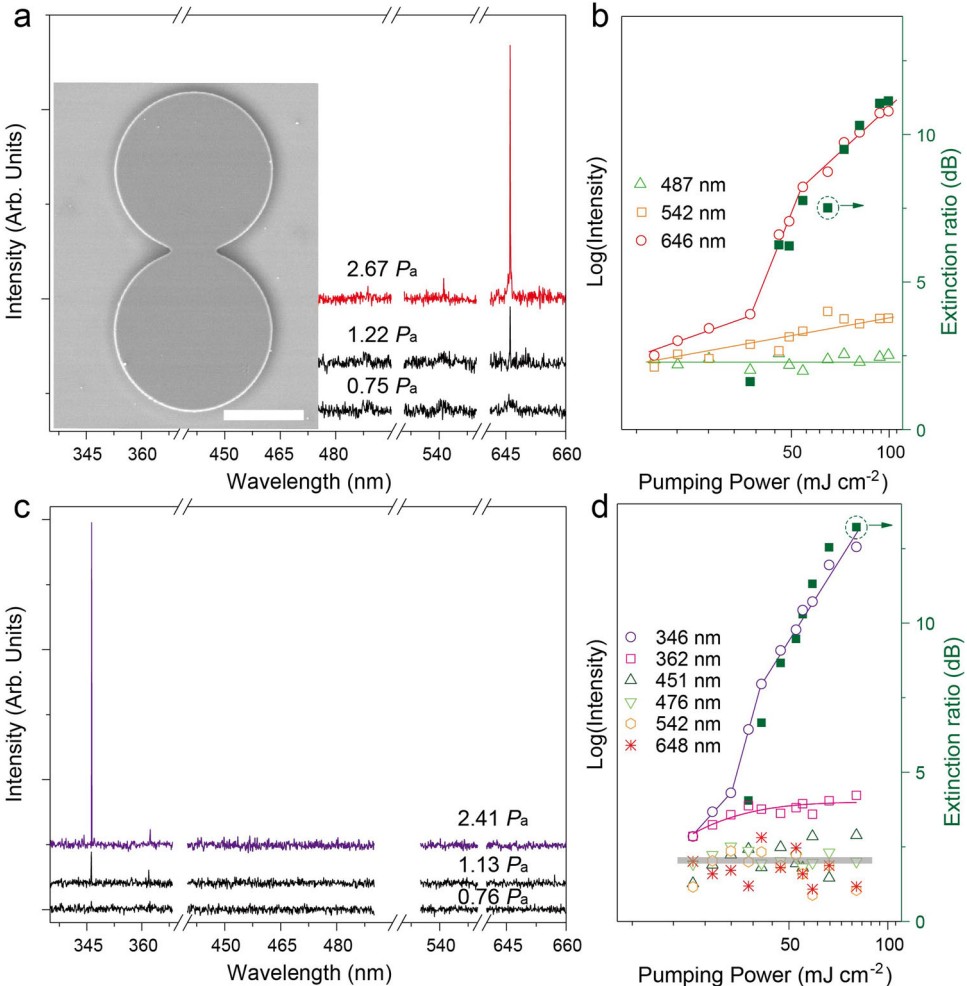

**Fig. 3 Realization of single-mode lasing in UCNCs-based size-mismatched PMs device.** The emission spectra of PMs structure are under (**a**) 808 nm, and (**c**) 980 nm right pumping, respectively. The inset gives the SEM image of a single PMs structure with a scale bar of 50 μm. The exact diameters of the top and down microdisks are 97.84 and 99.30 μm, respectively. **b**, **d** Characteristic light-light curves of the typical peaks in (**a**, **c**), respectively. The green filled boxes represent the E-ratio, and solid lines are linear fittings to guide the eyes.

linewidth is found to be around 0.05 nm at a pumping power similar to $P_{th}$, corresponding to a quality(Q)-factor of around 6900. Additionally, the transition from spontaneous emission through amplification to lasing oscillation, reflected by three regions with distinct slopes, is clearly visible from the corresponding light-light curves[2,3,12,31–33]. Both this power-dependent behavior and the well-defined mode spacing values confirm the lasing action along WGMs. Similar multiwavelength multimode WGMs lasing behavior has also been observed in other characteristic peaks (Section 3 in Supporting Information), except for the resonance peaks at 542 nm and 648 nm under the excitation of 980 nm laser. This is because those emissions cannot provide sufficient gain to support population inversion. It should be noticed that a multimode WGMs lasing at around 646.2 nm with the $P_{th}$ at around 60.99 mJ cm$^{-2}$ can be observed from the microdisk cavity under the excitation of 808 nm laser. As anticipated, the emission profiles of the observed multiwavelength lasers (see the solid lines in Fig. 2e) exhibit significant differences when excited by 808 nm and 980 nm lasers, respectively.

The WGMs lasing action derived from $Ln^{3+}$-doped materials has been well understood in many similar systems[2,31–33]. Notably, single-mode lasers can be realized in precisely regulated resonators with the diameter down to 4 μm or 20 μm[2,38]. Moreover, an enhanced 290 nm (i.e., $^1I_6 \rightarrow {}^3H_6$ of $Tm^{3+}$) lasing was

reported in a microdisk laser by keeping the thickness of UCNCs-based microdisk at a "cutoff" value (around 130–140 nm) to suppress competitive emission at 345 nm (i.e., $^1I_6 \rightarrow {}^3F_4$ of $Tm^{3+}$)[32]. However, other characteristic peaks, including 362 nm, 451 nm, 476 nm, 545 nm, and 648 nm of $Tm^{3+}$ ions, can hardly be simultaneously quenched at a given cutoff thickness or by other chemical approaches. Consequently, such multiwavelength, multimode, and isotropic output features in $Ln^{3+}$-based microdisk lasers lead to temporal and spatial fluctuations of the light source[39,40], thus are destructive to spectral purity and beam quality.

For precise mode management, we show that by harnessing notions of parity-time symmetry system[41–43], the dynamically tunable single-mode laser can be readily implemented in size-mismatched UCNCs-based PMs (inset of Fig. 3a) by asymmetric pumping and external excitation modulation. Such PMs structure was proceeded by standard photolithography, followed by a second spin-coating round using a polymer compound containing UCNCs (6.5 wt%). With the mature CMOS technique, the size and shape of each microdisk can be well repeated in the system of two adjacent circular cavities, except for a slight size-mismatching effect arising from fabrication inaccuracies of the photolithography technique. Here, we define such PMs lasers as two tangent microdisks with similar diameters of each

component under asymmetric excitation, which can be experimentally realized through selectively pumping at only one constituent resonator by a NIR pulsed laser.

As shown in Fig. 3a under 808 nm right pumping, a broad luminescent peak at around 646 nm (i.e., $^5F_5 \rightarrow {}^5I_8$ transition of $Ho^{3+}$ ions) emerges at the very beginning with the full width at half-maximum (FWHM) of ~8 nm. With the increase of pumping power, the spectra become quite different. Only one peak at 646.2 nm ascends from the emission band and grows rapidly above $P_{th}$ (i.e., 54.23 mJ cm$^{-2}$). The linewidth of the single mode action, once pumped above $P_{th}$, is less than ~0.07 nm, corresponding to a quality (Q)-factor of ~9000 (i.e., $Q = \lambda/\delta\lambda$, where $\lambda$ and $\delta\lambda$ denote the resonance peak and its FWHM, respectively). Figure 3b displays the corresponding light-light curves of three characteristic peaks of $Ho^{3+}$ ions (i.e., 486, 543, and 646 nm, respectively). The integrated intensity of the obtained mode as a function of power density (red open circles in Fig. 3b) presents a clear S-curve, which unambiguously confirms the onset of single-mode lasing at 646.2 nm[2,3,12,31–33]. In this regard, we introduce the extinction ratio (E-ratio), defined as $10\log(I_1/I_2)$ (where $I_1$ and $I_2$ are the intensity of the dominant peak and the highest side one, respectively), to estimate the performance of the supermode. It is observed that the corresponding E-ratio value (Fig. 3b) ascends steeply and reaches a value as high as 11 dB under 808 nm right pumping. Also, PMs laser exhibits a violet single-mode operation at the wavelength of 345.6 nm (i.e., $^1I_6 \rightarrow {}^3F_4$ transition of $Tm^{3+}$ ions) with an E-ratio exceeding 13 dB under 980 nm right pumping (Fig. 3c, d). The corresponding $P_{th}$ value is found to be around 41.23 mJ cm$^{-2}$. Once above $P_{th}$, the linewidth of spontaneous emission shrinks to less than 0.04 nm. Hence, our PMs device retains a single-mode laser in a wide pumping range with switchable wavelengths from 345.6 to 646.2 nm.

It should be noticed that the threshold values of the single-mode laser in our PMs device are 41.23 mJ cm$^{-2}$@345.6 nm and 54.23 mJ cm$^{-2}$@646.2 nm, respectively, which are obviously lower than that of the microdisk lasers (i.e., 65.68 mJ cm$^{-2}$@345.8 nm, and 60.99 mJ cm$^{-2}$@646.2 nm). In principle, the PMs structure under asymmetric excitation holds more losses compared with a simple microdisk laser. Nonetheless, owing to the unique features related to the metastable energy levels of $Ln^{3+}$ ions, a significant enhancement of the resulting supermode can be achieved. According to Fig. 3b, d, there are no obvious kinks from the light-light curves of other competing peaks (i.e., at 487 and 542 nm of $Ho^{3+}$ ions, and 362, 451, 476, and 648 nm of $Tm^{3+}$ ions). Those peaks thus act as spontaneous emission in our selectively excited PMs structure. It is generally accepted that the lifetime of the spontaneous emission is around hundreds of milliseconds, while that of the lasing emission is on the order of picoseconds, which implies a rapid depopulation rate[32]. Thus, as the onset of single-mode lasing, it will rapidly exhaust most energy at the selected level. Namely, the incident energy from the suppressed modes will directly contribute to the improvement of the resulting single-mode lasing.

The observed mode-selection phenomenon can be understood with the Vernier effect, which is caused by the detuning ($\Delta$)-dependent destructive interference[4,5,12,14,44,45]. In principle, the magnified free spectral range ($FSR_{enlarged}$) of the Vernier envelope[46,47] can be expressed as a function of the FSRs of the two coupled microdisks

$$FSR_{enlarged} = \frac{FSR_{right} \times FSR_{left}}{FSR_{right} - FSR_{left}} \quad (2)$$

where $FSR_{right}$ and $FSR_{left}$ are the FSRs of the two tangent microdisks. For the case of $d_{left} = 100 \ \mu m$, the diameter difference between two microdisks ($\Delta d$) is as small as 80 nm to realize the

designed spectral range (300 nm). Such a small difference imposes severe challenge in fabrication. More importantly, the threshold change caused by the mode coupling is proportional to the square of the detuning, $\Delta^2$ [47]. The tiny difference between two microdisks makes the detuning extremely small, and thus the threshold difference between the consecutive maxima of the Vernier envelope and its neighboring mode is negligible. Consequently, even one can realize the very large spectral envelope by the Vernier effect, it still faces severe challenges of realizing single-mode operation.

To overcome the obstacle of Vernier effect, a unique strategy encompassing the benefits of the mode coupling and the interaction with matter has been proposed. Here, the $\Delta d$ value is significantly increased from 80 nm to 1.46 μm (see the inset in Fig. 3a). Such a large size difference can be simply achieved by standard photolithography technique. Meanwhile, $\Delta$ has also been enhanced by a factor of 18 times. Of course, $FSR_{enlarged}$ is now reduced to 16.53 nm. Nonetheless, this value can be further improved by tailoring the gain spectrum[18,19,24–28] of the UCNCs. Namely, together with the Vernier effect, such UCNCs can be reversely designed to increase the spectral range.

Figure 4a gives the schematic design of the proposed PMs device, where the designed dual-mode upconversion material is purposely incorporated with the PMs structure. Note that two microdisks directly contact each other without any spacing. We numerically calculated two sets of resonance with identical gain spectrum to demonstrate mode-selection and mode-switching phenomena. In the calculation, $d_{left}$, $d_{right}$, and $n_{eff}$ are fixed at 5.28 μm, 6.762 μm, and 1.58, respectively. The matter of consideration here is that each constituent peak of the multicolor spectrum shares different FSRs owing to the intrinsic property of $Ln^{3+}$ ions with plentiful energy states in the violet and visible regions. Following the Vernier effect (Fig. 4b), once PMs is activated under NIR right excitation, most of the modes leak into the passive cavity through the joint area except for the selected modes, which satisfy the following equation: $mFSR_{left} = nFSR_{right}$ (i.e., where m and n should be co-prime integers)[4,5,12,14,44,45].

The mode crossing, indicated by the green dotted line in Fig. 4b, occurs periodically in the spectrum ranging from 330 to 700 nm. The threshold of resulting high-Q supermode will be significantly lower than their neighboring mode pairs[46,47]. This envelope of Vernier effect is smaller the desirable spectral range (i.e., 300 nm). However, the reversely designed UCNC with selected $Ln^{3+}$ is able to tailor the electronic states and introduce gain only to the designed wavelengths. The other consecutive maxima of the Vernier envelope are left without gain. The enlarged drawings of the selected gain regions, determined by the characteristic transitions of the selected activator ions, are shown in Fig. 4b. In this study, $Tm^{3+}$ and $Ho^{3+}$ are selected for purpose of filtering out the undesired Vernier modes. Obviously, by tailoring the emission profiles of the gain medium, most of the unwanted modes can be effectively eliminated from the resulting spectra except for the ones at the wavelengths of ~346 nm and ~646 nm. Namely, benefiting from the narrow bandwidth of the emission peaks and the abundant energy levels of $Ln^{3+}$ ions[24–28], the FSR can be readily enlarged up to 300 nm[4,6–23].

The switching between two single-mode lasing requires additional design on the UCNCs. Guided by this principle, two distinct $Nd^{3+} \rightarrow Yb^{3+} \rightarrow Ho^{3+}$ and $Yb^{3+} \rightarrow Tm^{3+}$ energy transfer paths are incorporated into the multi-shell nanoarchitectures (The details are described in the fabrication section). By virtue of structural engineering, one of two mutually exclusive emission peaks at the wavelengths of 646 nm and 346 nm emerge from the dual-mode spectra. Controlled in a switch-like manner, our UCNCs-doped PMs device will provide outstanding mode-switching performance.

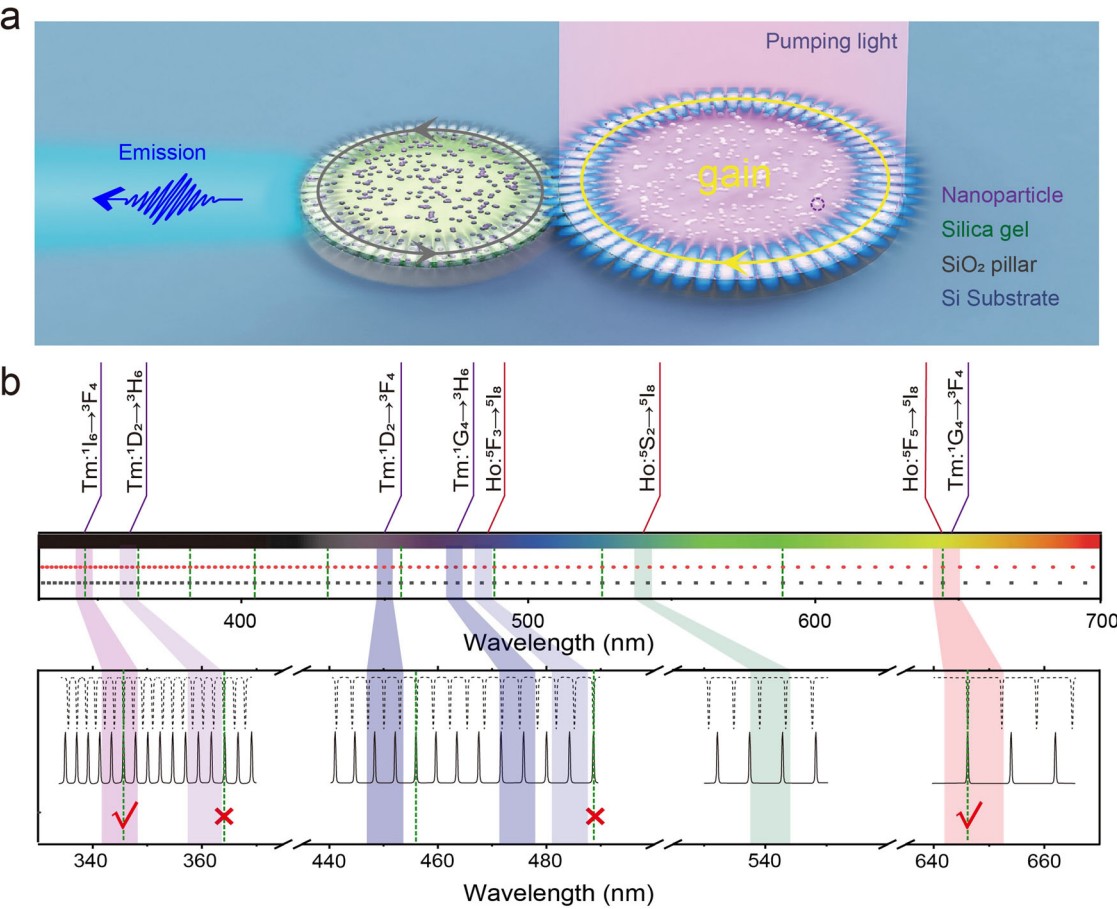

**Fig. 4 Theoretical calculation of the mode-switching mechanism in UCNCs-based size-mismatched PMs structure. a** Schematic illustration of the PMs structure. **b** The theoretical calculation (i.e., $d_{left} = 5.28\,\mu m$, $d_{right} = 6.762\,\mu m$, and $n_{eff} = 1.58$) and the enlarged drawings, which is determined by the gain region of $Tm^{3+}$ and $Ho^{3+}$ ions (the highlighted area). The green dotted lines indicate the selected modes from the calculation.

A more rigorous analysis was conducted by systematically investigating the lasing action in the PMs under three types of configurations, including left pumping, right pumping, and uniform pumping (Fig. S8). The corresponding simulated field distribution patterns are illustrated in the insets of Fig. S9a. Also, Fig. S9a,c give the normalized lasing spectra above $P_{th}$ under three types of pumping configurations with the corresponding light-light curves in Fig. S9b,d. The wavelengths (Fig. S10) of the selected modes slightly changes between the left and the right excited PMs. More interestingly, compared with the PMs device under NIR left pumping, the threshold value of the case under right pumping is lower (Figs. S9b and S9d) even though the two adjacent microdisks share nearly identical size and UCNCs-doping concentration. Both results confirm the size gap between the two coupled microdisks. Nonetheless, observation of the control data (Figs. S10 and S11) bolsters this narrative: the recorded spectra exhibit stable single-mode lasing emission while only one of the constituent resonators is excited, whereas the emission spectra show multimode lasing in uniformly pumped PMs. This result is understandable since there is an overlap of signals owing to selected modes from right pumping and left pumping, as well as the particular modes emerging from the coincidence of the resonance between left and right resonators under uniform pumping.

The uniformity and stability of these supermodes were further investigated in the short-listed PMs array (inset of Fig. 5). Despite the fabrication fluctuation of cavity size, a clear mode-switching phenomenon can be observed in all those neighboring PMs

(Fig. 5a, b), except for slight variance in the wavelengths of the selected modes and the corresponding thresholds (Fig. 5c, d, and S12). Also, from the repetitive emission switching by excitation cycling between 980 nm and 808 nm in Fig. 5e, this PMs laser switch shows excellent stability even with a certain amount of intensity distinctions of the supermode. Furthermore, this PMs structure can be rotated to attain the far-field pattern (Fig. S13), which quantitatively reveals that our single-mode laser emits along the direction of $\Phi FF = 180°$ with the divergence angle at around 30° under NIR right pumping. Such unidirectional emission directly results from the collimation of the unexcited cavity, while the emission in the opposite direction is suppressed by absorption[41], which is consistent with the simulation result for the emission at 646.2 nm (Fig. S13b). Obviously, this unidirectional single-mode laser favors the subsequent integration into on-chip photonic circuits in the corresponding direction, showing remarkable potential in the field of ultra-compact integrated optics. All these results present compelling evidence that our proposed strategy shows great advantages in pursuing switchable unidirectional single-mode lasing in submillimeter cavities with high tolerance in fabrication deviation and low demand for design complexity.

## Discussion

In summary, these findings demonstrate that our UCNCs-based size-mismatched PMs structure holds the promise for switchable unidirectional single-mode operation under asymmetric pumping

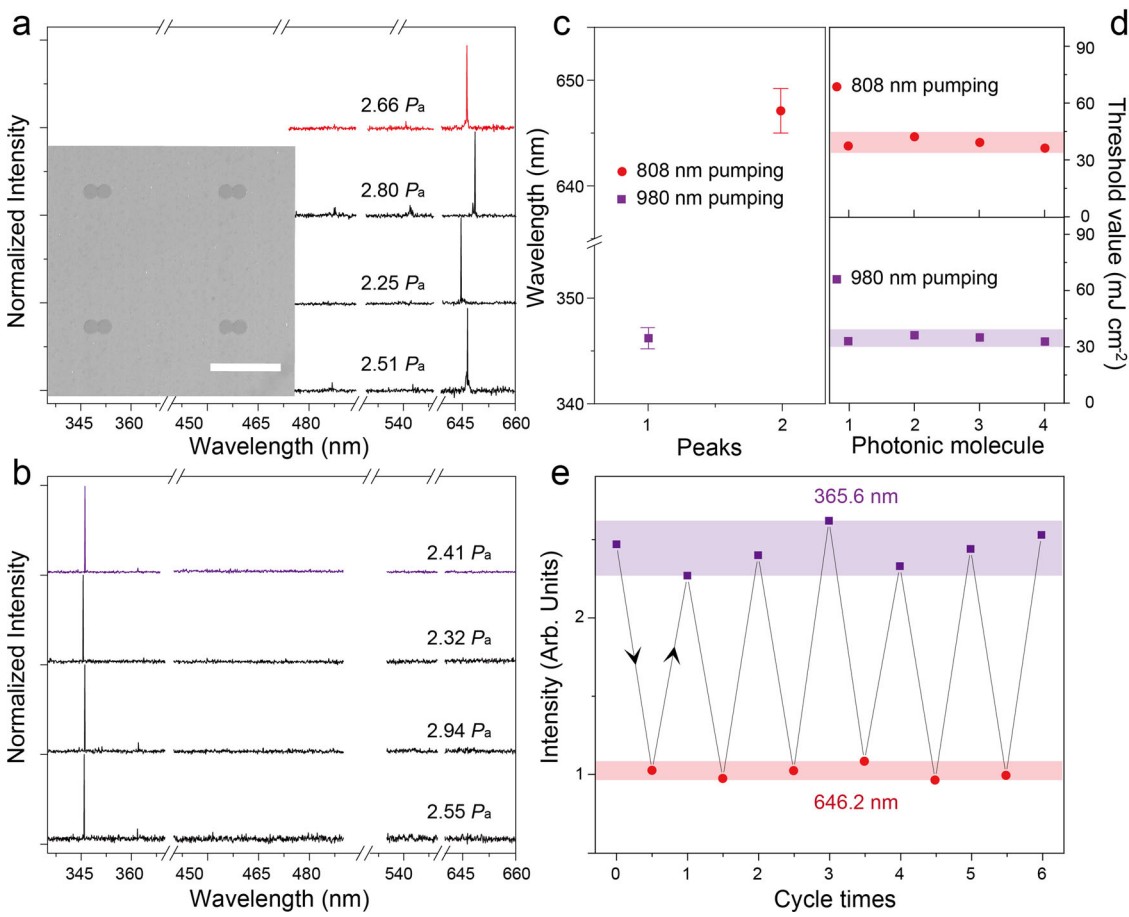

**Fig. 5 Uniformity of UCNCs-based PMs lasing.** Normalized emission spectra of the short-listed PMs devices under (**a**) 808 nm, and (**b**) 980 nm right pumping, respectively. The inset shows the SEM image of the PMs array with a scale bar of 500 μm. **c** The statistical deviation in wavelengths, and (**d**) thresholds of single-mode laser in (**a**, **b**). Each point in (**c**) is a mean value of four emission wavelengths, and the error bars stand for the s.d.'s from four sets of measurements. **e** The plot of the emission intensity of 365.6 and 646.2 nm against the 980–808 nm excitation cycles from the right excited PMs laser. The highlighted regions in Figure (**d**, **e**) are intended to guide the eyes.

and external excitation manipulation, even with the cavity size in the submillimeter scale. Such light modulation, including mode-selection, wavelength-switching, and emitting direction, is accomplished through favorable interactions between the gain and loss cavity in terms of the Vernier effect in conjunction with the manipulation of energy transfer in UCNCs. Apparently, many competing modes fall along with the amplification of selected modes in this homogeneous PMs device, resulting in an enhanced single-mode laser with a high contrast E-ratio up to 11 dB. The key idea behind this arrangement lies in the reversely designed UCNCs and the coupled microcavities. Indeed, we have proven excellent system given their strong light-tailoring and light-matter coupling that lead to a powerful mode-switching response in the form of single-mode lasing action. Particularly, this laser switch presents high uniformity, long-term stability, and an extremely wide spectral range (i.e., up to 300 nm). The investigation present substantial conceptual advance in the understanding of Ln³⁺-based coupled cavities. This strategy can be readily extended to other gain mediums, which undoubtedly raises new possibilities for UCNCs to complement or extend their applications from imaging to integrated photonic circuits.

## Methods

**Nanoparticle synthesis.** The multi-shell NaGdF₄:Yb/Nd@NaGdF₄:Yb/Ho@N-aYF₄:Ca@ NaYbF₄:Tm@NaYF₄:Ca nanocrystals were synthesized according to the method in ref.[2] Detailed fabrication and characterization are provided in the Supplementary Information.

**Device fabrication and characterization.** The proposed UCNCs-based microdisk laser and UCNCs-based PMs laser were implemented by tailoring the substrate[32]. The proposed UCNCs-based microdisk array was fabricated by a standard photolithography process (SVC model H94-25C) and the subsequent spin-coating operation. The UCNCs-based microcavities with a thickness of ~300 nm can be formed and characterized under the excitation of 808 or 980 nm pulsed lasers (6 ns, 10 Hz, Φ8mm), respectively. They are from a tunable laser system consisting of a 355 nm Q-switched Nd:YAG laser (Continuum, Surelite II-10) and an optical parametric oscillator (Continuum, Horizon I). The emission spectra were collected from the boundary of the photonic device by an optical fiber coupled to the iHR-320 (Horiba) monochromator (attached with a photomultiplier tube). The spectral resolution of spectrometer is 0.06 nm.

## Data availability

The supporting data generated in this study is publicly available in the online version of the paper.  Correspondence and request for materials should be addressed to L.J. or X.C.

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

## Acknowledgements

This work was supported by the NSFC of Guangdong Province (2018A030310034), NSFC (61805058, 51802198, 11974092, 12025402, 61975041, 11934012), Shenzhen Fundamental Research Fund (JCYJ20180306171700036, JCYJ20190806143813064, GXWD20201230155427003-20200821203750001, JCYJ20180507183532343, JCYJ20180507184613841, JCYJ20200109112805990, JCYJ20200109113003946, JCYJ20210324120402006), Fundamental Research Funds for the Central Universities. The authors also acknowledge support from the Shenzhen Engineering Laboratory on Organic-Inorganic Perovskite Devices, Shenzhen Scientific Research Foundation for the introduction of talent.

## Author contributions

L.J. conceived and supervised the optical experiments and analyzed the data. X.C. and H.S. designed and fabricated the luminescent materials. L.J. and X.C. wrote the paper. Y.W., X. Y., and L.J. fabricated the device and performed the optical characterization. X.C., L.J., Q.S., and S.X. contributed to analyzing and preparing this manuscript.

## Competing interests

The authors declare no competing interests.
