## [Peer review file · Nature Communications]

REVIEWER COMMENTS

Reviewer #1 (Remarks to the Author):

The authors described a type of dual-wavelength switchable single-mode lasers constructed by integrating coupled microdisk cavities with lanthanide-doped nanocrystals. Under excitation of different wavelengths, the lanthanide-doped nanocrystals supported lasing actions in two spectral bands (340-480 nm and 480-650 nm). By introducing coupled cavities for mode selection, they realized single-mode laser operation at ~345 and ~645 nm. Considering there is no scientific or technological breakthroughs in this work, I can not recommend the publication of this manuscript in Nature Communications.

1. Coupled microcavities are a well-known platform for realizing single-mode lasing based on diverse physical mechanisms, such as Vernier effect [Nano Lett. 2011, 11, 1122; Proc. Natl. Acad. Sci. USA 2013, 110, 865; Adv. Opt. Mater. 2014, 2, 220] and parity-time symmetry breaking [Science 2014, 346, 975; Laser Photonics Rev. 2016, 10, 494; Nat. Commun. 2017, 8, 15389]. Besides, the microfabrication technique of such microcavities have also been reported before [Adv. Mater. 2019, 31, 1807079].

2. Integration of multiple gain material components is the most commonly used method to obtain multicolor lasers [J. Am. Chem. Soc. 2012, 134, 12394; Nat. Nanotechnol. 2015, 10, 796; Nat. Photonics 2012, 6, 621; Sci. Adv. 2017, 3, e1700225]. The gain materials components include inorganic semiconductors and organic dyes. In this work, the authors merely replaced the semiconductors and dyes with lanthanide doped nanocrystals that were synthesized through a mature technology [Nat. Nanotechnol. 2015, 10, 924; Chem. Rev. 2017, 117, 4488].

Apart from the lack of novelty and significance, there are several other issues in this manuscript as follows.

1. There is no clear highlight shown in this manuscript. It seems that the laser with high performance (i.e. multicolor and high spectral purity) is what this manuscript tried to highlight. However, the authors used only two sentences to discuss the multicolor and single-mode lasers, respectively, in the Introduction section. Only one reference [Ref. 3, Nat. Nanotechnol. 2015, 10, 796] about the multicolor lasers and even no reference about the single-mode lasers was cited in the Introduction section. On the contrary, two whole paragraphs were written to discuss the luminescence of rare earth ions, in which most discussions have nothing to do with this work.

2. Many experimental and theoretical parameters were not given in this manuscript, which are necessary for the readers to understand this work. First, the authors claimed that the coupled

microdisks have different sizes due to the fabrication inaccuracy, but didn't give their exact diameter values. Second, the microdisk size and refractive index were not given for the theoretical calculation.

3. Statistical analyses of laser characteristics, such as threshold and linewidth/quality factor, should be given in the paper.

4. The Vernier effect can help understand the laser mode selection, but can't explain why the single-mode operation only took place at ~345 and ~645 nm. Actually, there are ~1000 modes (calculated by dividing spectral range of ~300 nm by mode spacing of ~0.24 nm) for each component microdisk in the coupled cavity system. I can't image that for so many modes, wavelength matching only occurs at the two gain regions around ~345 and ~645 nm rather than the others around ~360, ~450, ~475, ~485, and ~540 nm.

5. There are too many grammar mistakes in this manuscript. A thorough check and revision of the manuscript is suggested.

Reviewer #2 (Remarks to the Author):

This work reports on the design and incorporation of Ln³⁺-doped upconverting nanomaterial and two size-mismatched coupled cavity that is capable of generating (for the first time, I believe) a reversible ultraviolet-to-red unidirectional single-mode laser under simple excitation manipulation, even with the Ln³⁺-based cavity size in sub-millimeter scale. The authors have performed many control experiments (Ln³⁺-doped upconverting nanoparticles(UCNPs) incorporated with single microdisk and two coupled microdisks system, under different pumping configuration, under λ_1 - λ_2 excitation cycles, etc.) as well as different simulations, which make their experimental findings of robust mode-switching phenomenon across the whole visible spectrum very convincing.

The result is very interesting since multiwavelength laser with high purity is what researchers diligently strive after and the smart design of the nanomaterial and lasing cavity are impressive. Considering the previously reported spectral management of UCNPs from the bottom-up strategies, the incorporation of UCNPs and microresonator is one of the most probable solution to reach high monochromatic output with high reversibility in a wide tuning range. The results of this work updates the progress in this direction. The novelty is very high. I recommend the publication of this manuscript in Nature Communications and have a few comments that the authors can take into account for improving the clarity of some parts of their study.

Specific points and questions:

1) On page 5, a NaGdF₄:Yb/Nd@NaGdF₄:Yb/Ho@NaYF₄:Ca@NaYbF₄:Tm@NaYF₄:Ca core-multishelled structure is claimed in the manuscript. How much Yb/Nd/Ho/Tm ions are there in the nanocrystals? This would seem like a critical detail for dual-mode photoluminescence, but I don't see them specified until some TEM Figures. Please include in the figure legends and synthesis methods.

2) For 808 nm pumped case, why Nd³⁺ and Ho³⁺ ions are separated into two neighboring layers? In other words, Yb³⁺ ions doped in these two layers seems unnecessary and detrimental to the crosstalk-free dual-mode emission. Since Yb³⁺-dopants may absorb 980 nm incident energy and then transfer to Ho³⁺ ions, resulting in characteristic peaks at green and red band.

3) Single-mode operation is claimed through a combination of the Vernier effect and two different sets of mode spacing between two coupled microdisks. Is the effect limited to UCNCs, or should this be observable in normal luminescent materials?

4) The doping concentration of UCNCs is only 6.5 wt % in the UCNCs-based coupled microdisks cavity. There are still a lot of rooms. After reading Refs at the end of SI, the reviewer found that the side-mode suppression ratio is only around 11 dB/13 dB, much lower than that of the reported results such as perovskite nanowires (23 dB), comb-like ZnO structure(20.5 dB), GaN coupled nanowires (15.6 dB). Will the higher doping concentration of gain medium be beneficial for the increase of suppression ratio?

5) Regarding the lasing experiment, the author demonstrated an enhanced single-mode lasing (Line 243, "only enhanced amplification of overlapped modes at 646.2 nm and 345.6 nm could be observed ...". Line 314, "resulting in an enhanced single-mode laser with a high contrast ratio up to 11 dB. "). However, the enhancement should not be governed by the Vernier effect in the resonance in the proposed Ln³⁺-based microresonator. The reviewer would expect that some additional effect (external mode coupling [Physical Review Letters, 105,053902(2010); Laser & Photonics Review, 11 (5), 1700052 (2017)]) at the emission wavelengths should contribute to the improvement of selected mode.

6) There is two minor mistakes: in Page 14, Ref[33] and [35], the citation format should be identical with others.

Reviewer #3 (Remarks to the Author):

The authors obtain a dual-wavelength switchable single-mode lasing from a lanthanide-doped resonator. During many years have been published many papers related with cavity resonators,

however the possibility of a single-mode laser in different wavelengths made the results of this paper extremely novel.

The physics process is based in upconversion processes under different excitation wavelengths (808 or 980 nm) in the well-known multishelled Na(Gd or Y)F₄ nanocrystals doped with Nd³⁺, Yb³⁺, Ho³⁺ and Tm³⁺ ions. This process is explained in Fig. S2, although the authors use the phrase “...proposed energy exchange interaction...”. Maybe, the mechanism upconversion process is based on multipolar energy transfer interaction among ions instead of exchange interaction.

Using the upconverted nanoparticles and a silica resin the authors have prepared microdisk arrays as resonators. The idea of using two coupled resonators with different excitation setup is very interesting and it has given excellent results. The results are very interesting for a multidisciplinary scientific community because it allows to obtain micro-laser action in different wavelengths.

Respect to the analysis of the results, the authors emphasize the repeatability of the data. In this way, it is of special interest the Fig. 5c, where a very stable value is obtained for the threshold. Moreover in Fig. 5e is shown the deviation of the laser intensity. In my opinion, the low deviation in the threshold value is surprising, which can be explained in basis to the quality manufacture of the samples.

However, I would require that the authors explain the Fig. S11 in order to understand the pattern of the laser action. Isn't a laser action expected around 360 degrees?

In definitely, I consider the paper well organized and the results are consistent and well analyzed and it could be published in the present form.

Reply to Reviewer#1:

We thank the reviewer for the very careful review and valuable comments. We also appreciate the recognition from the reviewer for the realization of ultra-broadband single-mode laser switching across 300 nm (from 345 nm to 645 nm). Following the reviewer's comments, we have carefully revised the manuscript accordingly. The detail changes can be seen the point-to-point response below.

Comment-1: Coupled microcavities are a well-known platform for realizing single-mode lasing based on diverse physical mechanisms, such as Vernier effect [Nano Lett. 2011, 11, 1122; Proc. Natl. Acad. Sci. USA 2013, 110, 865; Adv. Opt. Mater. 2014, 2, 220] and parity-time symmetry breaking [Science 2014, 346, 975; Laser Photonics Rev. 2016, 10, 494; Nat. Commun. 2017, 8, 15389]. Besides, the microfabrication technique of such microcavities have also been reported before [Adv. Mater. 2019, 31, 1807079].

Our Response: We thank the reviewer for this valuable comment. We agree with the reviewer that the integration of the gain medium and the coupled microcavities can be found in many systems for the realization of single-mode laser on the basis of either Vernier effect or parity-time symmetry breaking mechanism [1-11]. However, we would like to point out that our mechanism is intrinsically different. **Our main goal is to realize the single-mode laser switching across a record large spectral range, e.g. $\Delta\lambda > 300$ nm.**

For the case of microdisk, its free spectral range (FSR) is mostly determined by its structural parameters, e.g., effective refractive index (n_{eff}) and diameter (d) following the equation [12-13]

$$FSR = \frac{c}{d n_{\text{eff}}} \quad (1)$$

For the microdisk in this research ($d = 100 \mu\text{m}$), the calculated FSR is 0.24 nm to 0.8 nm, orders of magnitude smaller than the typical gain spectral range. To enlarge the spectral range, the magnification of Vernier effect is naturally considered. Vernier effect is caused by the detuning (Δ)-dependent destructive interference. In principle, the magnified free spectral range (FSR_{enlarged}) of the Vernier envelope [12-13] can be expressed as a function of the $FSRs$ of the two coupled microdisks

$$FSR_{\text{enlarged}} = \frac{\lambda_1 \lambda_2}{\delta} \left(\frac{FSR_{\text{right}}}{FSR_{\text{right}}} - \frac{FSR_{\text{left}}}{FSR_{\text{left}}} \right) \quad (2)$$

Here $\lambda_{1,2}$ are the wavelengths of neighboring maximums of the envelope of Vernier effect and δ is the difference in optical paths of two cavities. FSR_{right} and FSR_{left} are the $FSRs$ of the two tangent microdisks. For the case of $d_{\text{left}} = 100 \mu\text{m}$, the diameter difference between two microdisks is as small as 80 nm to realize the designed

spectral range (300 nm).

Such a small difference imposes severe challenge in fabrication, which is almost impossible for the standard photolithography. Most importantly, the Vernier effect only consider the envelope and simply ignores the difference in laser thresholds. Basically, the threshold change caused by the mode coupling is proportional to the square of the detuning, Δ^2 [13]. The tiny difference between two microdisks makes the detuning Δ extremely small and thus the threshold difference between the consecutive maxima of the Vernier envelope and its neighboring mode is negligible. Consequently, even the Vernier effect can realize the very large spectral envelope, it still faces the challenge of realizing the single-mode operation.

To overcome the obstacle of Vernier effect, an unique strategy encompassing the benefits of the mode coupling and the interaction with matter has been proposed. We still utilize the Vernier effect but significantly increase the size difference from 80 nm to $\sim 1.46 \mu\text{m}$ (see Fig. R1). Such a large size difference can be simply achieved with standard photolithography. Meanwhile, the detuning Δ has also been enhanced by a factor of 18 times. Of course, the wavelength difference between consecutive maxima of the Vernier envelope is now reduced to 16.53 nm. This value, however, can be further improved by tailoring the gain materials [14-19]. In our research, the gain material is Ln^{3+} -based upconversion nanocrystals (UCNCs), which can be reversely designed to increase the spectral range.

Figure R1 (see the insets in Figure 3a and Figure 5a in the manuscript) The exact diameters of the coupled microdisks from the SEM images. The scale bars are 500 and 50 μm , respectively. The measured diameters of the right and left cavities are found to be 99.3 μm and 97.84 μm , respectively.

One example is schematically depicted in Fig. R2. The coupling between resonances in two resonators can form the the Vernier effect, where the mode crossing occurs periodically. While this envelope of Vernier effect is smaller the desirable spectral range (300 nm), the reversely designed of doped Ln^{3+} is able to tailor the electronic states and introduce gain only to the designed wavelengths. The other consecutive maxima of the Vernier envelope are left without gain. The enlarged drawings of the

selected gain regions, which are determined by the characteristic transitions of the selected activator ions, are shown in Fig. R2. Obviously, by tailoring the emission profiles of the gain medium, most of the unwanted modes can be effectively eliminated from the resulting spectra except for the ones at the wavelengths of 346 nm and 646 nm. Namely, benefiting from the narrow bandwidth of the emission peaks and the abundant energy levels of Ln^{3+} ions [14-19], the FSR can be readily enlarged up to 300 nm. This is one of the fundamental bases for the realization of single-mode laser-switch in our work. Therefore, while our work is similar to the previous report, the simultaneous consideration of optical states and electronic states makes it intrinsically different.

Figure R2 The enlarged drawings of the selected areas defined by the gain spectra of the selected Ln^{3+} ions. The characteristic transitions at the certain wavelengths are also attached for the guidance. For a clear view, the mode number for the Vernier effect has been reduced. All the overlapped modes are indicated by the green dotted lines.

In revised manuscript, we have modified the statement in **Page 9-11, Line 238-257, and 269-281**.

“The observed mode-switching effect can be understood with the Vernier effect, which is caused by the detuning (A)-dependent destructive interference. In principle, the magnified free spectral range (FSR_{enlarged}) of the Vernier envelope can be expressed as a function of the $FSRs$ of the two coupled microdisks

$$F S R_{\text{enlarged}} = \frac{FSR_{\text{right}} \times FSR_{\text{left}}}{|FSR_{\text{right}} - FSR_{\text{left}}|} \quad (2)$$

where FSR_{right} and FSR_{left} are the $FSRs$ of the two tangent microdisks. For the case of $d_{\text{left}} = 100 \mu\text{m}$, the diameter difference between two microdisks (Ad) is as small as 80 nm to realize the designed spectral range (300 nm). Such a small difference imposes severe challenge in fabrication. More importantly, the threshold change caused by the mode coupling is proportional to the square of the detuning, A^2 . The tiny difference between two microdisks makes the detuning extremely small, and thus the threshold difference between the consecutive maxima of the Vernier envelope and its neighboring mode is negligible. Consequently, even one can realize the very large spectral envelope by the Vernier effect, it still faces severe challenges of realizing single-mode operation.

To overcome the obstacle of Vernier effect, a unique strategy encompassing the benefits of the mode coupling and the interaction with matter has been proposed. Here, the Ad value is significantly increased from 80 nm to 1.46 μm (see the inset in Figure 3a). Such a large size difference can be simply achieved by standard photolithography technique. Meanwhile, A has also been enhanced by a factor of 18 times. Of course, FSR_{enlarged} is now reduced to 16.53 nm. Nonetheless, this value can be further improved by tailoring the gain spectrum of the UCNCs. Namely, together with the Vernier effect, such UCNCs can be reversely designed to increase the spectral range.”

“The mode crossing, indicated by the green dotted line in Figure 4b, occurs periodically in the spectrum ranging from 330 nm to 700 nm. The threshold of resulting high-Q supermodes will be significantly lower than their neighboring mode pairs. This envelope of Vernier effect is smaller the desirable spectral range (i.e., 300 nm). However, the reversely designed of doped Ln^{3+} is able to tailor the electronic states and introduce gain only to the designed wavelengths. The other consecutive maxima of the Vernier envelope are left without gain. The enlarged drawings of the selected gain regions, determined by the characteristic transitions of the selected activator ions, are shown in Figure 4b. In this study, Tm^{3+} and Ho^{3+} are selected for purpose of filtering out the undesired Vernier modes. Obviously, by tailoring the emission profiles of the gain medium, most of the unwanted modes can be effectively eliminated from the resulting spectra except for the ones at the wavelengths of 346 nm and 646 nm. Namely, benefiting from the narrow bandwidth of the emission peaks and the abundant energy levels of Ln^{3+} ions, the FSR can be readily enlarged up to 300 nm.”

Comment-2: Integration of multiple gain material components is the most commonly used method to obtain multicolor lasers. The gain materials components include inorganic semiconductors and organic dyes. In this work, the authors merely replaced the semiconductors and dyes with lanthanide doped nanocrystals that were synthesized through a mature technology [Nat. Nanotechnol. 2015, 10, 924; Chem. Rev. 2017, 117, 4488].

Our response: We really appreciate this valuable comment raised by the reviewer. We agree with the reviewer that different gain components can be integrated to realize the multiwavelength lasers. This strategy has been well explored in J. Am. Chem. Soc. 2012, 134, 12394 [20]; Nat. Nanotechnol. 2015, 10 796 [21]; Nat. Photonics 2012, 6, 621 [22]; Sci. Adv. 2017, 3, e1700225 [5], which have been cited as **Refs. 3, 8, 10, and 13** in the revised manuscript. For this technique, different gain materials are usually integrated in different locations. Each material has its own resonant modes. While the mode interaction can also be introduced, the difference between their refractive indices shall introduce additional difficulties in device design. More strikingly, the spatial division multiplexing of different colors inherently requires more space and will restrict the further reduction of device footprint. In our work, the gain spectra are controlled by the doped Ln^{3+} . Considering the desirable wavelengths, Tm^{3+} and Ho^{3+} are selected as the activator ions for purpose of filtering out the undesired Vernier modes (see Fig. R3 (b)). Since the ions are doped within the same nanoparticles, this strategy is suitable for the realization of multiwavelength lasers at any device sizes.

As mentioned above, the combination of Vernier effect and the control of gain spectral range can produce two single-mode lasing modes across a record large spectral range. The switching between two single-mode lasers requires additional design on the UCNCs. In our work, we further introduce a Nd^{3+} - Yb^{3+} sensitizer-pair, which enables different PL spectra of the same UCNCs under the excitation of 980 and 808 nm laser, respectively (see red and violet lines in Fig. R3).

Figure R3 The designed photoluminescence of $\text{Ho}^{3+}/\text{Tm}^{3+}$ codoped UCNCs. (a) Simplified

energy levels of Ho^{3+} and Tm^{3+} ions for the observed light in (b). (b) Theoretical calculation of the coupled microdisks system (i.e., $d_{\text{left}} = 5.28 \mu\text{m}$, $d_{\text{right}} = 6.762 \mu\text{m}$, and $n_{\text{eff}} = 1.58$) accompanying with the emission spectra from $\text{Ho}^{3+}/\text{Tm}^{3+}$ -codoped UCNCs with/without structural engineering, respectively. The pumping power is kept at 20 W cm^{-2} .

The detailed design principle of the multi-shell nanoarchitecture is described as follows (see Fig. R4). Two groups of energy transferring paths including Nd^{3+} - Yb^{3+} - Ho^{3+} and Yb^{3+} - Tm^{3+} are incorporated into the multi-shell $\text{NaGdF}_4:\text{Yb}/\text{Nd}(40/40)@\text{NaGdF}_4:\text{Yb}/\text{Ho}(48/2)@\text{NaYF}_4:\text{Ca}@\text{NaYbF}_4:\text{Tm}(1)@\text{NaYF}_4:\text{Ca}$ UCNCs. The $\text{Nd}^{3+}/\text{Yb}^{3+}$ sensitizer-pair clearly leads to two groups of characteristic peaks of Ho^{3+} and Tm^{3+} ions while being optically pumped at the wavelengths of 808 nm and 980 nm, respectively. Note that, the $\text{NaYF}_4:\text{Ca}$ inner layer with a thickness of 5 nm is used to prohibit the Nd^{3+} - Yb^{3+} - Tm^{3+} energy-mediated upconversion in the 808 nm laser-excited UCNCs, as well as the undesired Yb^{3+} - Ho^{3+} interfacial energy transfer in the 980 nm laser-excited case. The outermost layer of $\text{NaYF}_4:\text{Ca}$ with a thickness of 4 nm is employed to suppress surface quenching and protect upconversion process [14]. In addition, Nd^{3+} and Ho^{3+} ions are dispersed in the separated layers in order to avoid deleterious energy transferring between Nd^{3+} and the activator ions, while Yb^{3+} clusters are conducted in the neighboring layer to mediate the Nd^{3+} -sensitized upconversion process [23-24]. Thus, an efficient emission from the Ho^{3+} transitions can be obtained under the excitation of 808 nm laser due to the spatially confined $\text{Nd}^{3+}/\text{Ho}^{3+}$ ions and the effective Nd^{3+} - Yb^{3+} energy transfer (i.e., as high as 70%, depending on the proportion of contributing Nd^{3+} among all Nd^{3+} ions) [23]. Moreover, together with the inert inner-shell, the dopant amount of Yb^{3+} ions in the multi-shell nanoarchitecture is optimized to suppress the emission of the Ho^{3+} transitions and of the Tm^{3+} transitions for the 980 nm laser-excited case, therefore leading to a crosstalk-free dual-mode emission profiles (see the red and violet spectra in Fig. R4) under the excitation of 808 nm and 980 nm lasers, respectively. Such a design in UCNCs distinguishes this work from the mature technology in Nat. Nanotechnol. 2015, 10, 924 [25]; Chem. Rev. 2017, 117, 4488 [26] (cited as Refs. 24 and 28).

Figure R4 (see Figure S1 in the manuscript) The design principle of UCNCs. Two groups

of energy transferring paths (i.e., $\text{Nd}^{3+}\text{-Yb}^{3+}\text{-Ho}^{3+}$ and $\text{Yb}^{3+}\text{-Tm}^{3+}$, respectively) are enabled.

Overall, based on the simultaneous design on the optical modes and the electronic states, we have experimentally achieved the single-mode laser switching with a record large spectral range. We strongly believe that our findings, the simultaneous investigation of strong light-tailoring and light-matters coupling in the coupled microdisks cavity system present substantial conceptual advance in our understanding of Ln^{3+} -based PMs system.

In revised manuscript, we have modified the statement in **Page 4-5, Line 97-116**.

“As a proof of concept, $\text{NaGdF}_4\text{:Yb/Nd(40/40)@NaGdF}_4\text{:Yb/Ho(48/2)@NaYF}_4\text{:Ca@NaYbF}_4\text{:Tm(1)@NaYF}_4\text{:Ca}$ core-multishell UCNCs were synthesized via a modified literature procedure. In the conceptual design (Figure S1 in Supporting Information), two groups of energy transfer paths including $\text{Nd}^{3+}\text{-Yb}^{3+}\text{-Ho}^{3+}$ and $\text{Yb}^{3+}\text{-Tm}^{3+}$ are incorporated into the multi-shell nanocrystals. Obviously, the $\text{Nd}^{3+}/\text{Yb}^{3+}$ sensitizer-pair leads to two groups of characteristic peaks of Ho^{3+} and Tm^{3+} ions while being optically pumped at the wavelengths of 808 nm and 980 nm, respectively. Note that the $\text{NaYF}_4\text{:Ca}$ inner layer with a thickness of 5 nm is used to restrain the $\text{Nd}^{3+}\text{-Yb}^{3+}\text{-Tm}^{3+}$ energy-mediated upconversion in the 808 nm laser-excited UCNCs. The outmost layer of $\text{NaYF}_4\text{:Ca}$ with a thickness of 4 nm is employed to suppress surface quenching and protect the upconversion process (Figures S1 and S2). In addition, Nd^{3+} and Ho^{3+} ions are dispersed in the separated layers to avoid deleterious energy transfer between Nd^{3+} and the activator ions, while Yb^{3+} clusters are conducted in the neighboring layer to mediate the Nd^{3+} -sensitized upconversion process. Thus, an efficient emission from the Ho^{3+} transitions can be obtained under the excitation of 808 nm laser due to the spatially confined $\text{Nd}^{3+}/\text{Ho}^{3+}$ ions and the effective $\text{Nd}^{3+}\text{-Yb}^{3+}$ energy transfer (i.e., up to 70%, depending on the proportion of contributing Nd^{3+} among all Nd^{3+} ions). Moreover, together with the inert inner-shell, the doping amount of Yb^{3+} ions in the multi-shell nanoarchitecture is optimized to minimize the emission of the Ho^{3+} transitions and enhance the emission of the Tm^{3+} transitions for the 980 nm laser-excited case, therefore leading to a crosstalk-free dual-mode emission profiles (see the dotted lines in Figure 1e) under the excitation of 808 nm and 980 nm lasers.”

Comment-3: There is no clear highlight shown in this manuscript. It seems that the laser with high performance (i.e. multicolor and high spectral purity) is what this manuscript tried to highlight. However, the authors used only two sentences to discuss the multicolor and single-mode lasers, respectively, in the Introduction section. Only one reference [Ref. 3, Nat. Nanotechnol. 2015, 10, 796] about the multicolor lasers and even no reference about the single-mode lasers was cited in the Introduction section. On the contrary, two whole paragraphs were written to discuss the luminescence of rare earth ions, in which most discussions have nothing to do with this work.

Our Response: We really appreciate the reviewer for the constructed comment. To highlight the importance of our developments, we have explained the advantages and application of the tunable multiwavelength laser with high spectral purity in the revised manuscript. Please see **Page 2, Line 40-64**.

“Such miniaturized lasers can be achieved by exploiting a wavelength-tunable gain medium, that is effectively coupled into a microcavity. Among them, free-standing semiconductor alloy nanowires have received considerable attention, where the nanowires serve as both the gain medium and optical cavity. For example, Ning et al. have demonstrated a dual red-to-green single-mode lasing in a monolithic CdSSe alloy nanowire with the wide-gap end rolled up to a q-like shape. This strategy by controlling the composition gradient of a single heterostructure was also adopted to construct CsPbCl_{3-3x}Br_{3x} perovskite nanowires, CdS_xSe_{1-x} nanoribbons and metal-organic framework microcrystals (MOFs). Besides, the coupled microcavities [4-5,11-14], including heterogeneously coupled organic nanowires, homogeneously coupled ZnO microrods, folded CdSe nanowires and two mutually coupled microrings have been substantiated to generate a wavelength-tunable single-mode lasing based on either the Vernier effect or the parity-time symmetry breaking mechanisms, respectively. Nonetheless, those approaches usually require precise manipulation at the nanoscale. The remote dynamic modulation method is complementary to the above approaches. For instance, the dependence of lasing wavelengths on external stimuli, involving temperature, electric field, refractive index environment, and chemical surroundings, have been reported in many luminescent material systems, containing perovskites, organic molecules, polymers and so on. However, due to the limited availability of the active gain regions, the wavelength tuning range that can be realized from a single gain medium is quite limited in practice.

Upconversion nanocrystals (UCNCs), featured with plentiful energy levels and tailorable upconversion process, show great potential for switchable microlasers. Particularly, such materials capable of tunable light spanning the full spectrum from ultraviolet to near-infrared (NIR) are of significantly technological importance. Despite those advances, success in developing switchable Ln³⁺-based chip-integrated single-mode laser is extremely limited by their multiwavelength and multimode outputs. Therefore, it is vital to explore a novel mechanistic strategy that can generate broadband switchable single-mode lasing in mass-manufactural cavities.”

Comment-4: Many experimental and theoretical parameters were not given in this manuscript, which are necessary for the readers to understand this work. First, the authors claimed that the coupled microdisks have different sizes due to the fabrication inaccuracy, but didn't give their exact diameter values. Second, the microdisk size and refractive index were not given for the theoretical calculation.

Our Response: We thank the reviewer for carefully reading our manuscript and this

valuable comment. According to the SEM image of the coupled microdisks (see Figure R2), the exact diameter values of the left and right microdisk are found to be approximate 97.84 and 99.3 μm , respectively. Please see **Page 9, Line 234**. “The exact diameters of the top and down microdisks are 97.84 and 99.30 μm , respectively.”

In the theoretical calculation, d_{right} and d_{left} are fixed at 6.762/5.28 μm , and n_{eff} is 1.58, respectively. We have specified the detailed information in the revised manuscript. Please see **Page 11, Line 291**. “The theoretical calculation (i.e., $d_{\text{left}} = 5.28 \mu\text{m}$, $d_{\text{right}} = 6.762 \mu\text{m}$, and $n_{\text{eff}} = 1.58$).”

Comment-5: Statistical analyses of laser characteristics, such as threshold and linewidth/quality factor, should be given in the paper.

Our Response: We wish to thank the reviewer for the critical comments. As suggested, we have included specific values in the revised manuscript. They are listed as follows:

(1) **Page 6, line 145 and 157.**

“For instance, several periodic sharp peaks centered at 345.8 nm emerge from the broad emission band above the lasing threshold (i.e., $P_{\text{th}} = 65.68 \text{ mJ cm}^{-2}$ @ 345.8 nm, as reflected by the second kink value in Figure S6b), and then quickly dominate the emission spectra with increasing power. The experimental FSR reads as 0.24 nm, which matches the calculated one from the equation (1) [31-34]. The linewidth is found to be around 0.05 nm at a pumping power similar to P_{th} , corresponding to a quality(Q)-factor of around 6900.

It should be noticed that a multimode WGMs lasing at around 646.2 nm with the P_{th} at around 60.99 mJ cm^{-2} can be observed from the microdisk cavity under the excitation of 808 nm laser.”

(2) **Page 8, line 218.**

“With the increase of pumping power, the spectra become quite different. Only one peak at 646.2 nm ascends from the emission band and grows rapidly above P_{th} (i.e., 54.23 mJ cm^{-2}). The linewidth of the single mode action, once pumped above P_{th} , is less than 0.07 nm, corresponding to a quality (Q)-factor of 9000 (i.e., $Q = \lambda/\delta\lambda$, where λ and $\delta\lambda$ denote the resonance peak and its FWHM, respectively).”

(3) **Page 8, line 211-217.**

“The corresponding P_{th} value is found to be around 41.23 mJ cm^{-2} . Once above P_{th} , the linewidth of spontaneous emission shrinks to less than 0.04 nm.

It should be noticed that the threshold values of the single-mode laser in our PM device are 41.23 mJ cm^{-2} @ 345.6 nm and 54.23 mJ cm^{-2} @ 646.2 nm, respectively, which are obviously lower than that of the microdisk lasers (i.e., 65.68 mJ cm^{-2} @ 345.8 nm, and 60.99 mJ cm^{-2} @ 646.2 nm).”

Comment-6: The Vernier effect can help understand the laser mode selection, but can't explain why the single-mode operation only took place at 345 and 645 nm. Actually, there are 1000 modes (calculated by dividing spectral range of 300 nm by mode spacing of 0.24 nm) for each component microdisk in the coupled cavity system. I can't image that for so many modes, wavelength matching only occurs at the two gain regions around 345 and 645 nm rather than the others around 360, 450, 475, 485, and 540 nm.

Our Response: We wish to thank the reviewer for the critical comments. This is what the main innovation points are in this study.

Typically, for the case of $d_{\text{left}} = 100 \mu\text{m}$, the diameter difference (Ad) between the two coupled microdisks is as small as 80 nm by the Vernier effect to realize the designed spectral range (300 nm) [12]. Such a small Ad , and therefore a tiny laser threshold difference between the consecutive maxima of the Vernier envelope and its neighboring mode (AP_{th}) [13] make it difficult to realize the single-mode operation with the wavelength spanning over a very large spectral range.

Here, we propose an unique strategy encompassing the benefits of the mode coupling and the interaction with matter. The Ad value is significantly increased up to 1.46 nm, which is 18 times of that in the conventional mechanism. As a result, the detuning A has also been enhanced by a factor of 18 times, whereas the wavelength difference between consecutive maxima of the Vernier envelope is now reduced to 16.53 nm. Note that, AP_{th} is proportional to A^2 [13]. In addition, the FSR can be further enlarged by tailoring the gain materials through determined by the reverse design of UCNCs according to the theoretical calculations in Figure R2.

In revised manuscript, we have modified the statement in **Page 9-10, Line 238-257, and 269-281.**

“The observed mode-switching effect can be understood with the Vernier effect, which is caused by the detuning (A)-dependent destructive interference. In principle, the magnified free spectral range (FSR_{enlarged}) of the Vernier envelope can be expressed as a function of the $FSRs$ of the two coupled microdisks

$$FSR_{\text{enlarged}} = \frac{FSR_{\text{right}} \cdot FSR_{\text{left}}}{FSR_{\text{right}} - FSR_{\text{left}}} \quad (2)$$

where FSR_{right} and FSR_{left} are the $FSRs$ of the two tangent microdisks. For the case of $d_{\text{left}} = 100 \mu\text{m}$, the diameter difference between two microdisks (Ad) is as small as 80 nm to realize the designed spectral range (300 nm). Such a small difference imposes severe challenge in fabrication. More importantly, the threshold change caused by the mode coupling is proportional to the square of the detuning, A^2 . The tiny difference between two microdisks makes the detuning extremely small, and thus the threshold difference between the consecutive maxima of the Vernier envelope

and its neighboring mode is negligible. Consequently, even one can realize the very large spectral envelope by the Vernier effect, it still faces severe challenges of realizing single-mode operation.

To overcome the obstacle of Vernier effect, a unique strategy encompassing the benefits of the mode coupling and the interaction with matter has been proposed. Here, the Δd value is significantly increased from 80 nm to 1.46 μm (see the inset in Figure 3a). Such a large size difference can be simply achieved by standard photolithography technique. Meanwhile, Δ has also been enhanced by a factor of 18 times. Of course, FSR_{enlarged} is now reduced to 16.53 nm. Nonetheless, this value can be further improved by tailoring the gain spectrum of the UCNCs. Namely, together with the Vernier effect, such UCNCs can be reversely designed to increase the spectral range.”

“The mode crossing, indicated by the green dotted line in Figure 4b, occurs periodically in the spectrum ranging from 330 nm to 700 nm. The threshold of resulting high-Q supermodes will be significantly lower than their neighboring mode pairs. This envelope of Vernier effect is smaller the desirable spectral range (i.e., 300 nm). However, the reversely designed of doped Ln^{3+} is able to tailor the electronic states and introduce gain only to the designed wavelengths. The other consecutive maxima of the Vernier envelope are left without gain. The enlarged drawings of the selected gain regions, determined by the characteristic transitions of the selected activator ions, are shown in Figure 4b. In this study, Tm^{3+} and Ho^{3+} are selected for purpose of filtering out the undesired Vernier modes. Obviously, by tailoring the emission profiles of the gain medium, most of the unwanted modes can be effectively eliminated from the resulting spectra except for the ones at the wavelengths of 346 nm and 646 nm. Namely, benefiting from the narrow bandwidth of the emission peaks and the abundant energy levels of Ln^{3+} ions, the FSR can be readily enlarged up to 300 nm.”

Comment-7: There are too many grammar mistakes in this manuscript. A thorough check and revision of the manuscript is suggested.

Our Response: We have asked a native English speaker to help us proofread the manuscript. We hope that the revised manuscript is now clear and smooth to read.

Reply to Reviewer#2:

We thank the reviewer for the very careful review and valuable comments. We also appreciate the recognition from the reviewer for the realization of ultra-broadband single-mode laser switching across 300 nm (from 345 nm to 645 nm). Following the reviewer's comments, we have carefully revised the manuscript accordingly. The detail changes can be seen the point-to-point response below.

Comment-1: On page 5, a NaGdF₄:Yb/Nd@NaGdF₄:Yb/Ho@NaYF₄:Ca@NaYbF₄:Tm@NaYF₄:Ca core-multishelled structure is claimed in the manuscript. How much Yb/Nd/Ho/Tm ions are there in the nanocrystals? This would seem like a critical detail for dual-mode photoluminescence, but I don't see them specified until some TEM Figures. Please include in the figure legends and synthesis methods.

Our Response: We appreciate the reviewer for the very careful review and valuable comment. As suggested, the optimized doping concentration of Yb³⁺/Nd³⁺/Ho³⁺/Tm³⁺ ions (i.e., NaGdF₄:Yb/Nd(40/40)@NaGdF₄:Yb/Ho(48/2)@NaYF₄:Ca@NaYbF₄:Tm(1)@NaYF₄:Ca) was specified in the revised manuscript. Please see **Page 4, Line 97**.

"As a proof of concept, NaGdF₄:Yb/Nd(40/40)@NaGdF₄:Yb/Ho(48/2)@NaYF₄:Ca@NaYbF₄:Tm(1)@NaYF₄:Ca core-multishell UCNCs were synthesized via a modified literature procedure."

Comment-2: For 808 nm pumped case, why Nd³⁺ and Ho³⁺ ions are separated into two neighboring layers? In other words, Yb³⁺ ions doped in these two layers seems unnecessary and detrimental to the crosstalk-free dual-mode emission. Since Yb³⁺-dopants may absorb 980 nm incident energy and then transfer to Ho³⁺ ions, resulting in characteristic peaks at green and red band.

Our Response: We may not explain clearly in the manuscript so that the reviewer misunderstood the design principle of the as-proposed multi-shell upconversion nanocrystals (UCNCs).

The first comment is about the design principle of Nd³⁺/Ho³⁺ dopants that dispersed in the different layers. As has been previously reported, the introduction of Nd³⁺ ions would lead to the deleterious energy-transferring between Nd³⁺ and the activator ions [23-24]. Obviously, such Nd³⁺-related quenching could be highly restricted by separating Nd³⁺ and Ho³⁺ ions into the neighboring layers [23-24]. Moreover, the efficiency of Nd³⁺-Yb³⁺ energy transfer is as high as 70% in many types of host materials. Thus, we have established a NaGdF₄:Yb/Nd(40/40) @NaGdF₄:Yb/Ho(48/2) core-shell structure in this study. The core is doped with Nd³⁺ and Yb³⁺ ions, and the first shell is doped with Yb³⁺ and Ho³⁺ ions. Following the typical energy-mediated up-conversion mechanism, this Nd³⁺-sensitized nanostructure ensures successive Nd³⁺-Yb³⁺-Ho³⁺ energy transfer through Yb-clusters in the neighboring layers [23-24]. Consequently, an efficient emission from the Ho³⁺ transitions can be obtained

due to the spatially confined $\text{Nd}^{3+}/\text{Ho}^{3+}$ ions and the effective $\text{Nd}^{3+}\text{-Yb}^{3+}$ energy transfer.

Figure R5 (Figure S3. in the Supplementary Information) The inert-layer thickness dependent PL spectra. Emission spectra of the UCNCs with different shell thickness of intra $\text{NaYF}_4:\text{Ca}$ intralayer under the excitation of 808 nm laser.

The second important comment is about Yb^{3+} -dopants. In the conceptual design (Fig. R4), two group of energy transferring paths, including $\text{Nd}^{3+}\text{-Yb}^{3+}\text{-Ho}^{3+}$ and $\text{Yb}^{3+}\text{-Tm}^{3+}$, were incorporated into the multi-shell nanocrystals. The former Nd^{3+} -sensitized process provides characteristic emissions from the Ho^{3+} transitions under 808 nm irradiation, whereas the latter Yb^{3+} -sensitized one supports the typical emissions from Tm^{3+} ions under 980 nm irradiation. We agree with the reviewer that upon 980 nm pumping, $\text{Yb}^{3+}\text{-Ho}^{3+}$ and $\text{Yb}^{3+}\text{-Tm}^{3+}$ energy transferring processes may simultaneously happen owing to the Yb^{3+} -clusters in the core, first shell and third shell of UCNCs. This is detrimental to the realization of a crosstalk-free dual-mode photoluminescence. However, this negative effect could be highly suppressed by the establishment of inert inner-shell and the optimization of Yb^{3+} concentration in the core and first shell layers. As can be seen in Fig. R5, the introduction of $\text{NaYF}_4:\text{Ca}$ inert inner-shell significantly precludes the $\text{Nd}^{3+}\text{-Yb}^{3+}\text{-Tm}^{3+}$ energy-mediated upconversion in the 808 nm laser-excited UCNCs, which implies that this arrangement also eliminates undesired $\text{Yb}^{3+}\text{-Ho}^{3+}$ interfacial energy transfer in the 980 nm laser-excited one. Moreover, benefiting from the optimized concentration of Yb -dopant in the multi-shell nanoarchitecture, the characteristic peaks from the Ho^{3+} transitions are highly suppressed in the 980 nm laser-excited case (see the violet PL spectrum in Fig. R6). Namely, the introduction of Yb^{3+} facilitate the generation of the dual-mode photoluminescence in the proposed UCNCs under external excitation manipulation. We have added the corresponding descriptions in the revised

manuscript, see **Page 4-5, Line 97-116**.

“As a proof of concept, NaGdF₄:Yb/Nd(40/40)@NaGdF₄:Yb/Ho(48/2)@NaYF₄:Ca@NaYbF₄:Tm(1)@NaYF₄:Ca core-multishell UCNCs were synthesized via a modified literature procedure [2]. In the conceptual design (Figure S1 in Supporting Information), two groups of energy transfer paths including Nd³⁺-Yb³⁺-Ho³⁺ and Yb³⁺-Tm³⁺ are incorporated into the multi-shell nanocrystals. Obviously, the Nd³⁺/Yb³⁺ sensitizer-pair leads to two groups of characteristic peaks of Ho³⁺ and Tm³⁺ ions while being optically pumped at the wavelengths of 808 nm and 980 nm, respectively. Note that the NaYF₄:Ca inner layer with a thickness of 5 nm is used to restrain the Nd³⁺-Yb³⁺-Tm³⁺ energy-mediated upconversion in the 808 nm laser-excited UCNCs. The outmost layer of NaYF₄:Ca with a thickness of 4 nm is employed to suppress surface quenching and protect the upconversion process (Figures S1 and S2). In addition, Nd³⁺ and Ho³⁺ ions are dispersed in the separated layers to avoid deleterious energy transfer between Nd³⁺ and the activator ions [35], while Yb³⁺ clusters are conducted in the neighboring layer to mediate the Nd³⁺-sensitized upconversion process [35-37]. Thus, an efficient emission from the Ho³⁺ transitions can be obtained under the excitation of 808 nm laser due to the spatially confined Nd³⁺/Ho³⁺ ions and the effective Nd³⁺-Yb³⁺ energy transfer (i.e., up to 70%, depending on the proportion of contributing Nd³⁺ among all Nd³⁺ ions) [35-37]. Moreover, together with the inert inner-shell, the doping amount of Yb³⁺ ions in the multi-shell nanoarchitecture is optimized to minimize the emission of the Ho³⁺ transitions and enhance the emission of the Tm³⁺ transitions for the 980 nm laser-excited case, therefore leading to a crosstalk-free dual-mode emission profiles (see the dotted lines in Figure 1e) under the excitation of 808 nm and 980 nm lasers.”

Figure R6 (Figure 2e in the manuscript) PL spectra from the UCNCs-based solution under the excitation of CW 808/980 nm lasers respectively. The pumping power is around 20 W cm⁻².

Comment-3: Single-mode operation is claimed through a combination of the Vernier effect and two different sets of mode spacing between two coupled microdisks. Is the effect limited to UCNCs, or should this be observable in normal luminescent materials?

Our Response: We thank the review for the valuable comment. The Vernier effect is known as follows: certain resonance modes will be obtained when fulfilling both resonance conditions of the two isolated whispering gallery cavities, while other modes will be depressed [7, 12-13]. This strategy significantly increases the *FSR* and single-mode operation becomes possible when the *FSR* exceeds the gain region of active material. Obviously, this effect can be found in many optical systems, including ZnO semiconductor nanowire, GaN nanowire, polymer microfiber, CsPbBr₃ perovskite nanowire, and organic nanowire [1-2,5-6,10]. Particularly, the remarkably narrow linewidth of the lasing spectrum further enhances the capability for optical filters, sensors, and frequency-division-multiplexing transmission systems [1-2,5-6,10].

Comment-4: The doping concentration of UCNCs is only 6.5 wt % in the UCNCs-based coupled microdisks cavity. There are still a lot of rooms. After reading Refs at the end of SI, the reviewer found that the side-mode suppression ratio is only around 11 dB/13 dB, much lower than that of the reported results such as perovskite nanowires (23 dB), comb-like ZnO structure(20.5 dB), GaN coupled nanowires (15.6 dB). Will the higher doping concentration of gain medium be beneficial for the increase of suppression ratio?

Our Response: We thank the reviewer for the enlightening comment. As can be seen in Fig. R7, the doping concentration of UCNCs in the coupled microcavities system is determined by assessing the balance of absorbance and emission intensity of a set of solution containing different concentration of UCNCs [16]. Although a low dopant concentration of UCNCs can give rise to effective multiwavelength upconversion emissions, a relative higher UCNCs concentration is needed to maximize the emission in the ultraviolet and visible region. However, the substantially higher UCNCs concentration (e.g., 10, 20, and 35%) significantly quenches the upconversion emission intensity, especially the emission at the wavelength of ~346 nm. Moreover, the increasing of UCNCs deteriorates *Q*-factor of whispering gallery cavity, which plays a negative role in the realization of single-mode lasers. Hence, the doping concentration of UCNCs was fixed at 6.5 wt% throughout the paper. We have added the data in the revised Supplementary Information (Figure S5) along with a short discussion. Please see **Page 5 in Supporting Information**.

“As can be seen in Figure S5, the optimized doping concentration of UCNCs in the PM device is determined by assessing the balance of the absorbance and the emission intensity of a set of solution containing varying concentrations of UCNCs [2]. Although a low doping concentration of UCNCs can give rise to effective multiwavelength upconversion emission, a relative higher UCNCs concentration is

needed to maximize the violet emission. However, the substantially higher UCNCs concentration (e.g., 10, 20, and 35%) significantly quenches the upconversion emission, especially at the wavelength of ~346 nm. Moreover, the increasing of UCNCs concentration deteriorates Q-factor of the whispering gallery cavity. Namely, it may play a negative role in the realization of single-mode lasing. Hence, the doping concentration of UCNCs was fixed at 6.5 wt% throughout the paper, unless otherwise specified.”

Figure R7 (Figure S5 in the revised Supporting Information) The optimized

concentration of UCNCs in the PMs system. The dependence of (a) PL intensity under the excitation of 980 nm laser, and (b) the absorbance on the concentration of UCNCs in the cyclohexane solution.

Comment-5: Regarding the lasing experiment, the author demonstrated an enhanced single-mode lasing (Line 243, “only enhanced amplification of overlapped modes at 646.2 nm and 345.6 nm could be observed ...”. Line 314, “resulting in an enhanced single-mode laser with a high contrast ratio up to 11 dB. ”). However, the enhancement should not be governed by the Vernier effect in the resonance in the proposed Ln³⁺-based microresonator. The reviewer would expect that some additional effect (external mode coupling [Physical Review Letters, 105,053902(2010); Laser & Photonics Review, 11 (5), 1700052 (2017)]) at the emission wavelengths should contribute to the improvement of selected mode.

Our Response: We receive the comment with many thanks. It should be pointed out that the gain enhancement for selected modes in this study highly depends on the intrinsic metastable levels and energy-transferring upconversion mechanism of Ln³⁺.

Generally, the multiwavelength, multimode lasing action (Figure R9b) can be observed in an excited UCNCs-based microdisk cavity through the energy transfer upconversion mechanism [14]. The situation is quite different in UCNCs-based size-mismatched coupled microdisks system under asymmetric NIR excitation (Figure

R9b). Following the Vernier effect (see the inset of Figure R9b), once PMs is activated under NIR right excitation, most of the modes leak into the passive cavity through the joint area except for the selected modes, which satisfy the following equation: $mFSR_{\text{left}} = nFSR_{\text{right}}$ (i.e., where m and n should be co-prime integers) [5-7,12-13]. As can be seen from Figure R3b, several overlapped modes should emerge from the resulting spectra. Moreover, by tailoring the emitting profiles of the gain medium, most of the selected modes can be effectively eliminated from the resulting spectra, except for the supermodes at the wavelengths of 346 nm and 646 nm, respectively. Hence, the FSR can be readily enlarged to 300 nm.

It should be noticed that the threshold values of the single-mode laser in our PMs device, recorded as $41.23 \text{ mJ cm}^{-2}@345.6 \text{ nm}$ and $54.23 \text{ mJ cm}^{-2}@646.2 \text{ nm}$, respectively, are obviously lower than that of microdisk lasers (i.e., $65.68 \text{ mJ cm}^{-2}@345.8 \text{ nm}$, and $60.99 \text{ mJ cm}^{-2}@646.2 \text{ nm}$). In principle, the PMs structure under asymmetric excitation holds more losses compared with a simple microdisk laser. Nonetheless, owing to the unique features related to the metastable energy levels of Ln^{3+} ions, a unique enhancement of the resulting supermodes could be achieved. According to Figure 3b and 3d, there are no obvious kinks from the light-light curves of other competing peaks (i.e., at 487 and 542 nm of Ho^{3+} ions, and 362, 451, 476, and 648 nm of Tm^{3+} ions), which thus act as typical spontaneous emission in our selectively excited PMs structure. It is generally accepted that the lifetime of the spontaneous emission is hundreds of milliseconds, while that of the lasing emission is on the order of picoseconds, which implies a rapid depopulation rate [14]. Thus, as the onset of single-mode lasing, it will rapidly exhaust most energy at the selected level. Namely, the incident energy from the suppressed modes will directly contribute to the enhancement of the resulting single-mode lasing (see Figure R9a).

According to the suggestion, we have modified the statement in the revised manuscript, see **Page 8, Line 235-248**.

“It should be noticed that the threshold values of the single-mode laser in our PM device are $41.23 \text{ mJ cm}^{-2}@345.6 \text{ nm}$ and $54.23 \text{ mJ cm}^{-2}@646.2 \text{ nm}$, respectively, which are obviously lower than that of the microdisk lasers (i.e., $65.68 \text{ mJ cm}^{-2}@345.8 \text{ nm}$, and $60.99 \text{ mJ cm}^{-2}@646.2 \text{ nm}$). In principle, the PM structure under asymmetric excitation holds more losses compared with a simple microdisk laser. Nonetheless, owing to the unique features related to the metastable energy levels of Ln^{3+} ions, a significant enhancement of the resulting supermode can be achieved. According to Figure 3b and 3d, there are no obvious kinks from the light-light curves of other competing peaks (i.e., at 487 and 542 nm of Ho^{3+} ions, and 362, 451, 476, and 648 nm of Tm^{3+} ions). Those peaks thus act as spontaneous emission in our selectively excited PM structure. It is generally accepted that the lifetime of the spontaneous emission is around hundreds of milliseconds, while that of the lasing emission is on the order of picoseconds, which implies a rapid depopulation rate [32]. Thus, as the onset of single-mode lasing, it will rapidly exhaust most energy at the selected level. Namely, the incident energy from the suppressed modes will directly contribute to the improvement of the resulting single-mode lasing.”

Figure R8 (Figure 2e and Figure 3c in the manuscript) The enhanced single-mode action at 345.6 nm. (a) Simplified energy transfer mechanism for the observed light. (b) The multiwavelength multimode emission spectra from a single UCNCs-based microdisk cavity (top) and the coupled microdisks system (down) above P_{th} . The insets plot the simulations of single microdisk ($d = 3 \mu\text{m}$) and two coupled microdisks ($d_{\text{left}} = 5.28 \mu\text{m}$, $d_{\text{right}} = 6.762 \mu\text{m}$), respectively.

Comment-6: There is two minor mistakes: in Page 14, Ref[33] and [35], the citation format should be identical with others.

Our Response: We thank the reviewer for the careful review and valuable suggestion. Accordingly, we have modified the indicated ones, see **Page 16, Line 448, and Page 17, Line 457.**

“[39] Ma, R. M. & Oulton, R. F. Applications of nanolasers. *Nat. Nanotechnol.* **14**, 12-22 (2019).”

[44] Li, F. *et al.* Single-mode lasing of CsPbBr₃ perovskite NWs enabled by the Vernier effect. *Nanoscale* **13**, 4432-4438 (2021).”

Reply to Reviewer#3:

We thank the reviewer for the very careful review and valuable comments. We also appreciate the recognition from the reviewer for the realization of ultra-broadband single-mode laser switching across 300 nm (from 345 nm to 645 nm). Following the reviewer's comments, we have carefully revised the manuscript accordingly. The detail changes can be seen the point-to-point response below.

Comment-1: The physics process is based in upconversion processes under different excitation wavelengths (808 or 980 nm) in the well-known multishelled Na(Gd or Y)F₄ nanocrystals doped with Nd³⁺, Yb³⁺, Ho³⁺ and Tm³⁺ ions. This process is explained in Fig. S2, although the authors use the phrase "...proposed energy exchange interaction...". Maybe, the mechanism upconversion process is based on multipolar energy transfer interaction among ions instead of exchange interaction.

Our Response: We appreciate the reviewer for this valuable comment. We agree with the reviewer that upconversion mechanism follows the sequential absorption of excitation energy by the sensitizer ions and energy transfer interaction between Ln³⁺ ions [24-29]. In the study, the Nd³⁺→Yb³⁺→Ho³⁺ energy transfer satisfies the well-known energy-mediated upconversion mechanism [23-24]. As suggested, we have amended the indicated sentence in a more rational fashion, see **Page 3, Section 2, Figure S1 in the Supporting Information**.

"The proposed energy-mediated upconversion mechanism in the multi-shell nanocrystals"

Comment-2: However, I would require that the authors explain the Fig. S11 in order to understand the pattern of the laser action. Isn't a laser action expected around 360 degrees?

Our Response: We receive the comment with many thanks. In this study, there is a strong interaction between the gain and loss microcavities. Note that, the unexcited microcavity not only serves as the spectral filter of the other one, but also acts as a conventional lens. Thus, a unidirectional emission along the direction of $\Phi_{FF} = 180^\circ$ is developed by the collimation of the loss cavity, while the emission in the opposite direction was suppressed by absorption [7]. This is consistent with the simulation result at the emission wavelength of 646.2 nm (Figure R10b).

As suggested, we have added the data in the revised manuscript (Figure S12) along with a short discussion.

In revised manuscript, we have added the **Figure S12 in the Supporting Information**, and modified the statement in the manuscript, see **Page 12, Line 318-321**.

"Such unidirectional emission directly results from the collimation of the unexcited cavity, while the emission in the opposite direction is suppressed by absorption, which is consistent with the simulation result for the emission wavelength of 646.2 nm (Figure S12b)."

Figure R9 (Figure S12 in the revised manuscript) The unidirectional emission. (a) Far field pattern of the experimental lasing action at 646.2/345.6 nm under 808/980 nm right pumping, respectively. (b) The corresponding simulation result with $d_{\text{left}} = 5.28 \mu\text{m}$, $d_{\text{right}} = 6.762 \mu\text{m}$, $\lambda_0 = 646.2 \text{ nm}$, and $n_{\text{eff}} = 1.58$, respectively.

Reference

- [1] Ta, V. D., Chen, R. & Sun, H. Coupled Polymer Microfiber Lasers for Single Mode Operation and Enhanced Refractive Index Sensing. *Adv. Opt. Mater.* **2**, 220-225 (2014).
- [0]Xiao, Y. *et al.* Single-nanowire single-mode laser. *Nano Lett.* **11**, 112-126 (2011).
- [2] Liu, W. *et al.* An integrated parity-time symmetric wavelength-tunable single-mode microring laser. *Nat. Commun.* **8**, 15389 (2017).
- [1]Feng, L. *et al.* Single-mode laser by parity-time symmetry breaking. *Science* **346**, 972-975 (2014).
- [3] Zhang, C. *et al.* Dual-color single-mode lasing in axially coupled organic nanowire resonators. *Sci. Adv.* **3**, 170-225 (2017).
- [4] Wang, Y. Y. *et al.* Lasing mode regulation and single-mode realization in ZnO whispering gallery microcavities by the Vernier effect. *Nanoscale* **8**, 166-319 (2016).
- [2]Zhang, N. *et al.* Quasi parity-time symmetric microdisk laser. *Laser Photonics Rev.* **11**, 152-170 (2017).
- [5] Hodaei, H. *et al.* Parity-time-symmetric microring lasers. *Science* **346**, 975-978 (2014).
- [3]Peng, B. *et al.* Parity-time-symmetric whispering-gallery microcavities. *Nat. Phys.* **10**, 394-398 (2014).
- [6] Li, F. *et al.* Single-mode lasing of CsPbBr₃ perovskite NWs enabled by the Vernier effect. *Nanoscale* **13**, 4432-4438 (2021).
- [7] Gao, H. *et al.* Cleaved-coupled nanowire lasers. *Proc. Natl. Acad. Sci. USA.* **110**, 865-869(2013).
- [8] Gomes, A. D., Bartelt, H. & Frazo, O. Optical Vernier Effect: Recent Advances and Developments. *Laser Photonics Rev.* 2000588 (2021).
- [9] Ge, L., Tureci, H. E. Inverse Vernier effect in coupled lasers. *Phys. Rev. A* **92**, 013840 (2015).
- [10]Jin, L. M. *et al.* Mass-manufactural lanthanide-based ultraviolet B microlasers. *Adv. Mater.* **31**, 79-180 (2019).
- [11]Chen, X. *et al.* Confining energy migration in upconversion nanoparticles towards deep ultraviolet lasing. *Nat. Commun.* **7**, 103-104 (2016).
- [0] Jin, L. M. *et al.* Enhancing Multiphoton Upconversion from NaYF₄:Yb/Tm@NaYF₄ Core-Shell Nanoparticles via the Use of Laser Cavity. *ACS Nano* **11**, 843-849 (2017).
- [1] Wang, T. *et al.* White-light whispering-gallery-mode lasing from lanthanide-doped upconversion NaYF₄ hexagonal microrods. *ACS Photon.* **4**, 1539-1543 (2017).
- [2] Hai, Z. *et al.* Amplified Spontaneous Emission and Lasing from Lanthanide-Doped Up-Conversion Nanocrystals. *ACS Nano* **7**, 11420-11426 (2013).
- [12]Xu, X. *et al.* Random lasing in Eu³⁺ doped borate glass-ceramic embedded with Ag nanoparticles under direct three-photon excitation. *Nanoscale* **7**, 46-50 (2015).
- [13]Xu, J. *et al.* Room-temperature dual-wavelength lasing from single-nanoribbon lateral heterostructures. *J. Am. Chem. Soc.* **134**, 12394-12397 (2012).
- [14]Ning, C. Z. *et al.* A monolithic white laser. *Nat. Nanotechnol.* **10**, 796-803

(2015).

[15] Cerdán, L. *et al.* FRET-assisted laser emission in colloidal suspensions of dye-doped latex nanoparticles. *Nat. Photonics* **6**, 621-626 (2012).

[16] Wang, Y. *et al.* Nd³⁺-Sensitized Upconversion Nanophosphors Efficient In Vivo Bioimaging Probes with Minimized Heating Effect. *ACS Nano* **7**, (2013).

[17] Zhou, B. *et al.* NIR II-responsive photon upconversion through energy migration in an ytterbium sublattice. *Nat. Photon.* **14**, 760-766 (2020).

[18] Zhou, B., Shi, B., Jin, D. & Liu, X. Controlling upconversion nanocrystals for emerging applications. *Nat. Nanotechnol.* **10**, 924–936 (2015).

[19] Qin, X., *et al.* Lanthanide-Activated Phosphors Based on 4f-5d Optical Transitions: Theoretical and Experimental Aspects. *Chem Rev.* **117**, 4488-4527 (2017).

[20] Wang, F. *et al.* Simultaneous phase and size control of upconversion nanocrystals through lanthanide doping. *Nature* **463**, 1061-1065 (2010).

[21] Zheng, K. *et al.* Recent advances in upconversion nanocrystals: Expanding the kaleidoscopic toolbox for emerging applications. *Nano Today* **29**, 100797 (2019).

[22] Wang, F. & Liu, X. Upconversion multicolor fine-tuning: visible to near-infrared emission from lanthanide-doped NaYF₄ nanoparticles. *J. Am. Chem. Soc.* **130**, 5642-5643 (2008).

REVIEWER COMMENTS

Reviewer #1 (Remarks to the Author):

I have carefully read the revised manuscript and the responses to the reviewers' comments. The authors indeed made substantial revisions according to the comments, which notably improve the quality of the manuscript. However, I am not convinced that there are any scientific or technological breakthroughs in this work. In terms of novelty and importance, this work is far from the high-quality research that Nature Communications looks for.

1. Mode selection mechanism

In my last review report, I questioned the novelty of the mode selection mechanism demonstrated in this work based on the fact that coupled microcavities are a well-known platform for realizing single-mode lasing by exploiting diverse physical mechanisms, such as Vernier effect (see the following Refs. 1-3) and parity-time (PT) symmetry breaking (Refs.4-6). In this work, the laser mode selection relies on the notion of PT-symmetry. Note that such a mode selection phenomenon in the size-mismatched microdisk cavity has been reported by the same research group, where they call it a quasi-PT symmetric microdisk laser (Ref.7).

In their response to my comment, the authors try to claim that their mode selection mechanism is intrinsically different from those previously reported mechanisms by confusing the mode selection and wavelength switching. Actually, the mode selection and wavelength switching dictate the laser resonant mode and output waveband, respectively. Although enabling a dual-wavelength switchable laser across a record large spectral range, the simple combination of the mode selection and wavelength switching does not reveal any new physical mechanism.

2. Multiwavelength output

It is necessary to iterate that the integration of multiple gain material components is the most commonly used method to obtain multicolor lasers (Refs.8-12). The gain materials components include inorganic semiconductors, organic dyes, and rare-earth ions. Hence, this work using lanthanide-doped nanocrystals does not provide a new materials design concept for multiwavelength lasers.

The authors state that "different gain materials are usually integrated in different locations", suffering from difficult device design and large device footprints. That's definitely not true because a lot of papers have reported the uniform integration of different gain materials (Refs.10, 12-18).

3. Fabrication technique

The microfabrication technique of the microdisk cavity lasers have been reported before (Ref.19).

4. Mode selection result

In my last review report, I expressed my great concern about why the single-mode operation only took place at ~345 and ~645 nm rather than ~360, ~450, ~475, ~485, and ~540 nm. Such a weird laser mode selection phenomenon can not be explained by the Vernier effect.

In their answer to my comment, the authors first state that a 300 nm mode spacing can be realized in the coupled microdisks with diameters of ~ 100 μm and a diameter difference of 80 nm. Even so, the mode matching is a random event in the coupled microdisks due to the inevitable microfabrication inaccuracy, and consequently the single-mode lasing is supposed to occur at a single wavelength of ~360, ~450, ~475, ~485, and ~540 nm or simultaneously at two wavelengths of ~345 and ~645 nm with equal probabilities. The theoretical probability of achieving the dual-wavelength single-mode lasing is only 1/6.

As was pointed out by the authors, the actual diameter difference is as large as 1.46 μm , which results in a 16.53 nm mode spacing. The much-narrowed mode spacing corresponds to higher probabilities of mode matching at ~360, ~450, ~475, ~485, and ~540 nm. As a result, the theoretical probability of achieving lasing at only ~345 and ~645 nm, as illustrated in Figure 4 and R2, dramatically decreases.

The authors further claim that the mode spacing “can be further improved by tailoring the gain materials”. Specifically, by tailoring the electronic states, optical gain can be introduced only to the desired ~345 and ~645 nm (Figure 4 and R2). Obviously, the authors attempt to play with language to increase the importance of this work. In fact, the optical gain regions of these rare-earth ions, that is, Ho³⁺ and Tm³⁺, are fixed (Figure 2e). What the authors does shown in Figure 4 and R2 is to calculate a small-probability mode matching at only ~345 and ~645 nm by tuning the cavity modes in theory. Even if assuming that the low-probability event did occur in the authors’ experiment, I still can not imagine how the authors realized such low-probability events four times in a 2×2 coupled microdisk array (Figure 5).

The final issue might be the most serious one. As shown in Figure 4, the radiative transitions of Tm³⁺ at ~360, ~450, ~475 nm occur from the energy levels (1D₂, 1D₂, 1G₄, and 5F₃, respectively) different from that (1I₆) at ~345 nm, which means there is no competition between the optical modes at ~360, ~450, ~475 nm and those at ~345 nm. The situation is the same in Ho³⁺. As shown in Figure 3 and 5, the highest pumping power of the coupled microdisks exceeds 100 mJ cm⁻², which is much higher than the thresholds (~ 60 mJ cm⁻²) of lasing at ~360, ~450, ~475, ~485, and ~540 nm. Because of the absence of mode competition, lasing is supposed to appear at ~360, ~450, ~475, ~485, and ~540 nm in the coupled microdisks. I have to express great concern about the validity of the data on the dual-wavelength single-mode lasing.

Overall, I recommend a rejection of this manuscript.

References

1. Xiao Y, et al. Single-nanowire single-mode laser. *Nano Lett* 11, 1122-1126 (2011).

2. Gao H, Fu A, Andrews SC, Yang P. Cleaved-coupled nanowire lasers. *Proc Natl Acad Sci USA* 110, 865-869 (2013).
3. Ta VD, Chen R, Sun H. Coupled Polymer Microfiber Lasers for Single Mode Operation and Enhanced Refractive Index Sensing. *Adv Opt Mater* 2, 220-225 (2014).
4. Hodaei H, Miri M-A, Heinrich M, Christodoulides DN, Khajavikhan M. Parity-time-symmetric microring lasers. *Science* 346, 975-978 (2014).
5. Hodaei H, et al. Single mode lasing in transversely multi-moded PT-symmetric microring resonators. *Laser Photonics Rev* 10, 494-499 (2016).
6. Liu W, et al. An integrated parity-time symmetric wavelength-tunable single-mode microring laser. *Nat Commun* 8, 15389 (2017).
7. Zhang N, et al. Quasiparity-Time Symmetric Microdisk Laser. *Laser Photonics Rev* 11, 1700052 (2017).
8. Xu J, et al. Room-Temperature Dual-Wavelength Lasing from Single-Nanoribbon Lateral Heterostructures. *J Am Chem Soc* 134, 12394-12397 (2012).
9. Fan F, Turkdogan S, Liu Z, Shelhammer D, Ning CZ. A monolithic white laser. *Nat Nanotechnol* 10, 796-803 (2015).
10. Cerdan L, et al. FRET-assisted laser emission in colloidal suspensions of dye-doped latex nanoparticles. *Nat Photonics* 6, 621-626 (2012).
11. Zhang C, et al. Dual-color single-mode lasing in axially coupled organic nanowire resonators. *Sci Adv* 3, e1700225 (2017).
12. Haider G, et al. A Highly-Efficient Single Segment White Random Laser. *ACS Nano*, (2018).
13. Galisteo-López JF, Ibasate M, López C. FRET-Tuned Resonant Random Lasing. *J Phys Chem C* 118, 9665-9669 (2014).
14. Ta VD, et al. Multicolor lasing prints. *Appl Phys Lett* 107, 221103 (2015).
15. Zhang Y, et al. Dual-wavelength lasing from organic dye encapsulated metal-organic framework microcrystals. *Chem Commun* 55, 3445-3448 (2019).
16. Tong J, et al. Dual-color plasmonic random lasers for speckle-free imaging. *Nanotechnology* 31, 465204 (2020).
17. Kamran MA, et al. Dual-Color Lasing Lines from EMPs in Diluted Magnetic Semiconductor CdS:NiI Structure. *Research* 2019, 6956937 (2019).
18. Gao Y, et al. Green Stimulated Emission Boosted by Nonradiative Resonant Energy Transfer from Blue Quantum Dots. *J Phys Chem Lett* 7, 2772-2778 (2016).
19. Jin L, et al. Mass-Manufactural Lanthanide-Based Ultraviolet B Microlasers. *Adv Mater* 31, e1807079 (2019).

Reviewer #2 (Remarks to the Author):

accept.

Reviewer #3 (Remarks to the Author):

The authors have answered the issues made by the referee. Therefore, this paper can be published in the present form.

Comments from Reviewer #1:

Comment-1: Mode selection mechanism. In my last review report, I questioned the novelty of the mode selection mechanism demonstrated in this work based on the fact that coupled microcavities are a well-known platform for realizing single-mode lasing by exploiting diverse physical mechanisms, such as Vernier effect (see the following Refs. 1-3) and parity-time (PT) symmetry breaking (Refs.4-6). In this work, the laser mode selection relies on the notion of PT-symmetry. Note that such a mode selection phenomenon in the size-mismatched microdisk cavity has been reported by the same research group, where they call it a quasi-PT symmetric microdisk laser (Ref.7).

In their response to my comment, the authors try to claim that their mode selection mechanism is intrinsically different from those previously reported mechanisms by confusing the mode selection and wavelength switching. Actually, the mode selection and wavelength switching dictate the laser resonant mode and output waveband, respectively. Although enabling a dual-wavelength switchable laser across a record large spectral range, the simple combination of the mode selection and wavelength switching does not reveal any new physical mechanism.

Our Response: We receive the comments from the reviewer with many thanks. According to reviewer #1's suggestion, we have carefully read the suggested references. It seems to us that reviewer #1 failed to understand the mode-selection behavior in this work. While these references relate to our research, there are some apparent differences.

1) The reference #1, "Nano Lett. 2011, 11, 1122", focuses on the double end-looped CdSe nanowire laser which follows the Vernier effect. A tunable single-mode nanowire laser with the emission wavelengths from 733.7 to 726.9 nm has been realized by changing the geometry of one constituent loop, leading to the reduction in optical path of the lasing cavity. This arrangement highly relies on the micromanipulation under the microscope. Both the emission wavelength and the tunable range are hard to be repeated in other devices. This is obviously different from our results in Fig. 5 of the main text and the additional results in Fig. RR2 below.

A similar Vernier effect can be observed in reference #3, "Adv. Opt. Mater., 2014, 2, 220", of which the basic concept is the size-mismatched coupled RhB-based polymer fibers. The emission wavelength is controlled by changing the refractive index of the surrounding solution and can be well repeated. However, the wavelength shift is only around 0.4 nm, orders of magnitude smaller than our observation.

The research in reference #2, "PNAS, 2013, 110, 865", reveals a single-frequency lasing from a cleaved-coupled Fabry-Perot cavity with the intracavity gap of around 60-nm. It is worth noting that the length ratio and the air gap between two component nanowires are critical for achieving single-mode lasing in this article. However, the emission wavelengths of single-mode lasing in this design cannot be switched by the Vernier effect.

2) In case of the reference #4, "Science, 2014, 346, 975", the selective breaking of parity-time (PT) symmetry (the introduction of loss part) clearly ensures a stable single-longitudinal mode operation from a coupled system of two identical InGaAsP rings with 200-nm air gap.

Similarly, the same group (reference #5, “Laser Photonics Rev. 2016, 10, 494”) presented the transition from multimode behavior in microring cavity to single-mode behavior in a transversely multimoded twin-ring system as enabled by preferential PT symmetry breaking. However, there is no switching of single-mode laser in these works.

With the assistance of phase modulator, Yao’s group (reference #6, “Nat. Commun. 2017, 8, 15389”) has extended the above concept and demonstrated an electrically pumped PT-symmetric twin-ring configuration that can support a continuously tunable single-mode laser emission. However, the emission wavelength can only be tuned from 1553.80 to 1554.02 nm, orders of magnitude smaller than our observation too.

3) The reference #7, “Laser Photonic Rev., 2017, 11, 1700052”, mainly focuses on the demonstration of the external coupling mechanism for single-mode lasing in the dye-doped size-mismatched double-microdisk cavity. The blue-shift of the lasing wavelength (~5 nm), that is also orders of magnitude smaller than our results, is caused by the thermal effect.

In our work, we have demonstrated the switching between two single-mode lasing actions separated by 300 nm. As mentioned in the manuscript, this is realized with the combined efforts of Vernier effect and the reverse design of UCNCs. The Vernier effect with relatively large size mismatching can enlarge the FSR and preserve the gain difference between the coupled resonances and the neighboring modes. While many possible Vernier modes may emerge from the spectrum, the unwanted ones cannot lase due to the control of the gain spectrum of the luminescent materials. Moreover, mode-switching behavior can be further achieved through an additional design on the UCNCs with two distinct $\text{Nd}^{3+} \rightarrow \text{Yb}^{3+} \rightarrow \text{Ho}^{3+}$ and $\text{Yb}^{3+} \rightarrow \text{Tm}^{3+}$ energy transfer paths being incorporated into the multi-shell nanoarchitecture. With the control of pumping laser, these two paths can be selectively excited and switch the single-mode lasing action between ~646 nm and ~346 nm. These performances are almost impossible to be realized with a single mechanism such as Vernier effect or PT symmetry breaking and thus simply distinguish our work from the references #1-6 mentioned by Reviewer #1.

Comment-2: Multiwavelength output. It is necessary to iterate that the integration of multiple gain material components is the most commonly used method to obtain multicolor lasers (Refs.8-12). The gain materials components include inorganic semiconductors, organic dyes, and rare-earth ions. Hence, this work using lanthanide-doped nanocrystals does not provide a new materials design concept for multiwavelength lasers.

The authors state that “different gain materials are usually integrated in different locations”, suffering from difficult device design and large device footprints. That’s definitely not true because a lot of papers have reported the uniform integration of different gain materials (Refs.10, 12-18).

Our Response: We thank the reviewer for the suggestions of these valuable references. We carefully read these references. Our research mainly focuses on the widely switchable lasing within a microlaser device from one single gain medium (i.e., the reversely designed UCNCs). The suggested Refs. #8-12 reveal multicolor lasers from the heterostructures or composites of different luminescent materials. We will discuss the detailed differences below.

The reference #8, “J. Am. Chem. Soc. 2012, 134, 12394”, demonstrates a dual-color lasing from a three-segment CdSSe nanoribbon lateral heterostructure. Similarly, a multi-segment ZnCdSSe nanoribbon heterostructure (Ref. 9, “Nat. Nanotechnol., 2015, 10, 796”) was fabricated to simultaneously generate all three elementary colors, and thus leading to the realization of white lasers. Notably, the reference # 11 (“Sci. Adv. 2017, 3, e1700225”) that proposed a switchable dual-color single-mode lasing is quite similar to our work. Nonetheless, this green-to-blue lasing were realized from two axially coupled organic heterogeneous nanowires. Obviously, these multicolor lasing arising from side-by-side FP cavities or axially coupled FP cavities are a light combination from different locations.

The research in reference #10, “Nat. Photonics, 2012, 6, 621”, demonstrated multicolor lasing from the Rh6G/NB colloidal mixture, while the reference #12, “ACS Nano 2018, 12, 11847”, reported hyperbolic meta-materials assisted white lasing from lanthanide-doped nanoparticles. However, the tuning of lasing wavelengths was realized by modifying the chemical composition of the gain medium that incorporated in the random cavities.

2) The statement of “For this technique, different gain materials are usually integrated in different locations” is true. For example, based on Förster resonance energy transfer (FRET) mechanism, the Refs. #10,13-14 and 18 have demonstrated multicolor lasing emission from the donor(D)/acceptor(A) mixtures (i.e., Refs. #10,13-14: two families of organic dyes, Refs. #18: II-VI semiconductor quantum dots). Generally, the emission wavelengths can be adjusted by changing the constituent scattering medias (i.e., either the gain medium with different D/A ratios (Refs. #10,13-14,18) or the random scatterers (Refs. #13)). The Refs. #15 and 16 have proposed dual-wavelength random lasers by encapsulating the composite of two luminescent materials (i.e., organic dyes, semiconductor quantum dots) as the gain medium. Interestingly, the research in Ref. #15, “Chem. Commun. 2019, 55, 3445”, supports the switchable dual-wavelength multimode lasing by changing the excitation wavelength. However, it is found that two lasing species in these achievements obviously share different closed feedback loops in random cavities.

As for the references #12 and 17, these two works focus on the multicolor lasing from one single gain medium rather than “the uniform integration of different gain materials”. Typically, the Chen group (Ref. #12) have reported an enhanced white random laser by using lanthanide-codoped nanocrystals as the gain medium. Besides, the NiI-doped CdS nanobelts (NBs) are proposed in Ref. #17, which supports dual-color lasing from excitonic magnetic polaron mechanism. Interestingly, this lasing behavior shows different power-dependence on the excitation power, thus the relative intensity of dual-color lasing could be tuned. But in fact, those two lasing peaks are not mutually exclusive. Hence, the output are a light combination of the emissions at around 530 nm and 789 nm, and it is quite difficult to obtain tunable lasing in this research. Namely, despite of the uniform integration of the gain medium, it cannot support tunable lasing in these designs.

Here, it should be pointed out that in this work Ln^{3+} ions are uniformly distributed in the crystal lattice of NaYF_4 host. Namely, such gain medium, that is uniformly integrated at

atomic level, can be regarded as one single luminescent material. Obviously, the stoichiometric amount of Ln^{3+} ions determines the gain spectrum of the UCNCs, which provides a unique possibility for widely switchable single-mode lasing.

Comment-3: Fabrication technique. The microfabrication technique of the microdisk cavity lasers have been reported before (Ref.19).

Our Response: The main contribution of this work is striving to take advantages of both the reversely designed UCNCs and the coupled cavities to realize precise mode management in microlaser, therefore enabling remarkable tunability and dynamic functionality. This strategy may extend the application of UCNCs to a new class of on-chip photonic devices. Therefore, the microfabrication technique itself is not the important part of this research.

Comment-4: Mode selection result. In my last review report, I expressed my great concern about why the single-mode operation only took place at ~ 345 and ~ 645 nm rather than ~ 360 , ~ 450 , ~ 475 , ~ 485 , and ~ 540 nm. Such a weird laser mode selection phenomenon can not be explained by the Vernier effect.

In their answer to my comment, the authors first state that a 300 nm mode spacing can be realized in the coupled microdisks with diameters of ~ 100 μm and a diameter difference of 80 nm. Even so, the mode matching is a random event in the coupled microdisks due to the inevitable microfabrication inaccuracy, and consequently the single-mode lasing is supposed to occur at a single wavelength of ~ 360 , ~ 450 , ~ 475 , ~ 485 , and ~ 540 nm or simultaneously at two wavelengths of ~ 345 and ~ 645 nm with equal probabilities. The theoretical probability of achieving the dual-wavelength single-mode lasing is only 1/6.

As was pointed out by the authors, the actual diameter difference is as large as 1.46 μm , which results in a 16.53 nm mode spacing. The much-narrowed mode spacing corresponds to higher probabilities of mode matching at ~ 360 , ~ 450 , ~ 475 , ~ 485 , and ~ 540 nm. As a result, the theoretical probability of achieving lasing at only ~ 345 and ~ 645 nm, as illustrated in Figure 4 and R2, dramatically decreases.

The authors further claim that the mode spacing “can be further improved by tailoring the gain materials”. Specifically, by tailoring the electronic states, optical gain can be introduced only to the desired ~ 345 and ~ 645 nm (Figure 4 and R2). Obviously, the authors attempt to play with language to increase the importance of this work. In fact, the optical gain regions of these rare-earth ions, that is, Ho^{3+} and Tm^{3+} , are fixed (Figure 2e). What the authors does shown in Figure 4 and R2 is to calculate a small-probability mode matching at only ~ 345 and ~ 645 nm by tuning the cavity modes in theory. Even if assuming that the low-probability event did occur in the authors’ experiment, I still can not imagine how the authors realized such low-probability events four times in a 2×2 coupled microdisk array (Figure 5).

The final issue might be the most serious one. As shown in Figure 4, the radiative transitions of Tm^{3+} at ~ 360 , ~ 450 , ~ 475 nm occur from the energy levels (1D₂, 1D₂, 1G₄, and 5F₃, respectively) different from that (1I₆) at ~ 345 nm, which means there is no competition between the optical modes at ~ 360 , ~ 450 , ~ 475 nm and those at ~ 345 nm. The situation is the same in Ho^{3+} . As shown in Figure 3 and 5, the highest pumping power of the coupled

microdisks exceeds 100 mJ cm^{-2} , which is much higher than the thresholds ($\sim 60 \text{ mJ cm}^{-2}$) of lasing at ~ 360 , ~ 450 , ~ 475 , ~ 485 , and $\sim 540 \text{ nm}$. Because of the absence of mode competition, lasing is supposed to appear at ~ 360 , ~ 450 , ~ 475 , ~ 485 , and $\sim 540 \text{ nm}$ in the coupled microdisks. I have to express great concern about the validity of the data on the dual-wavelength single-mode lasing.

Our Response: We appreciate the reviewer for these very valuable comments. We fully agree with the Reviewer #1 that the mode matching randomly occurs in the coupled microdisks. However, there is a relatively fixed processing accuracy of standard photolithography technique. The size difference between two microdisks are fixed within a small range. In this sense, while the exact wavelengths of the single-mode lasing vary in different samples, their emission spectral range are well preserved at ~ 346 and $\sim 646 \text{ nm}$. This can be seen from the results in Fig. 5a-c of the main text. There are indeed some other Vernier modes at the wavelengths of ~ 360 , ~ 377 , ~ 396 , ~ 416 , ~ 438 , ~ 463 , ~ 491 , ~ 522 , ~ 557 , and $\sim 597 \text{ nm}$ (indicated by the dotted lines in Fig. RR1), respectively. Fig. RR1 shows the normalized amplified spontaneous emission (ASE) and the photoluminescence spectra of the designed UCNCs. Apparently, a sharp reduction in the emission linewidths can be observed with the transition from PL emission to ASE. Due to the material design, the reversely designed UCNCs with sharp emission lines have much lower gain in these spectral ranges (even though the PL peaks cover such Vernier modes) and thus no lasing emissions can be observed.

Figure RR1: The normalized emission spectra (gray ones) and ASE spectra (colored ones) under the excitation of 980/808 nm laser, respectively. The exact wavelengths of Vernier modes are indicated by the dotted lines.

Following the reviewer's comment, we have fabricated another series of sample to confirm the device reproducibility. The results are shown in Fig. RR2 below. We can see the emission

wavelengths of single-mode lasers are slightly different in nine neighboring PM structures due to the fabrication inaccuracy. But their lasing spectral ranges well fall around 646 nm and 346 nm. These observations are consistent with the results in the main text even though the samples are fabricated in different times. Note that the large aggregates of UCNCs (marked by the orange arrows in <2> and <6>) at the boundary of PM devices would significantly reduce the E-ratio of the resulting supermode. This is because the spoil of Q-factor would lead to the enhancement of spontaneous emission around the other peaks (i.e., ~362, ~451, ~476, ~487, and ~542 nm). But the dominant modes are still the ones at ~646 nm and ~346 nm. These results clearly show the good reproducibility of our devices.

Figure RR2: The short-listed PM devices. (a) The optical images of another 3×3 array (marked from <1> to <9>), and each PM structure in the array. The surface roughness of each microcavity can be deduced from the thickness observed along the central axis of PM device (see the white dotted line in <1>). The normalized lasing spectra of the short-listed PM devices under the optical excitation at (b) 980 nm, and (b) 808 nm, respectively. The pumping fluence is around 85 mJ cm⁻².

Another important comment is about tailoring the gain profiles of UCNCs. It is true that the energy level diagrams of these Ln³⁺ ions are fixed and usually exhibit multi-peak emissions owing to their abundant levels of energy states (Fig. RR3(a) and top panel in Fig. RR3(b)). However, the optical gain of the Ln³⁺-doped UCNCs are not fixed. The gain from radiative recombination of different excited states can be selectively enhanced or quenched by controlling the doping concentration of Ln³⁺ ions and structural engineering of UCNCs. With

these approaches, the designed $\text{Tm}^{3+}/\text{Ho}^{3+}$ -codoped UCNCs have maximal gains at around 346 nm and 646 nm under the excitation of 808 nm lasers (see top panel in Fig. RR3(b)). Such kind of gain control is mature and have been widely reported in literatures. In our research, we have further designed the core-multishell UCNC structure with two distinct energy transfer paths. By inserting an inert shell between them, two types of gain paths can be constructed and they can be selectively excited under different excitation conditions.

With such kind of material engineering, the dominant gain happens at 330-480 nm bands under laser excitation at 980 nm (see bottom panel in Fig. RR3(b)). When the pumping switches to 808 nm, the gain also switches to 480-660 nm bands (see middle panel in Fig. RR3(b)). In this sense, while there are many Vernier modes as mentioned by the reviewer, the engineering on gain spectra makes the device lase at ~ 646 nm and ~ 346 nm only. This is also well consistent with the repeated observations in Fig. RR2 and Fig. 5 in the main text.

Figure RR3 (Figure R3): The photoluminescence of $\text{Ho}^{3+}/\text{Tm}^{3+}$ co-doped UCNCs. (a) Simplified energy levels of Ho^{3+} and Tm^{3+} ions for the observed light in (b). (b) The emission spectra from $\text{Ho}^{3+}/\text{Tm}^{3+}$ -co-doped UCNCs with/without structural engineering, respectively. The pumping power is kept at 20 W cm^{-2} .

Reviewer #2 (Remarks to the Author):
accept.

Reviewer #3 (Remarks to the Author):
The authors have answered the issues made by the referee. Therefore, this paper can be published in the present form.

Our response: We thank the two referees for their careful review and suggestions. We also appreciate that both reviewers are satisfied with the novelty of this work.

REVIEWERS' COMMENTS

Reviewer #1 (Remarks to the Author):

The revised version of the manuscript is now ready for publication.

Comment 1: Reviewer #1 (Remarks to the Author):

The revised version of the manuscript is now ready for publication.

Our response: We appreciate for the referee's recommendations. We also would like to express our gratitude for his/her time and useful comments.

Comment 2: Abstract and editor's summary

Your paper will be accompanied by the following editor's summary. Please let us know if there are any inaccuracies: 'Dual wavelength lasers are gaining importance for photonic applications. In this work the authors demonstrate that by coupling a lanthanide material to a two-size mismatched coupled microcavities, it is possible to achieve laser switch with high uniformity, long-term stability, and an extremely wide spectral range up to 300nm'

Our response: We really appreciate for your time and effort. The short summary is modified as follows: “Dual-wavelength lasers are gaining importance for photonic applications. In this work the authors demonstrate that by incorporating a lanthanide-doped material into the size-mismatched coupled microcavities, it is possible to achieve laser switch with high uniformity, long-term stability, and an extremely wide spectral range up to 300nm.”

Comment 3: Author information

Please review your complete author list to verify that it is complete and accurate. We ask that you consult with your coauthors to ensure that all names, affiliations, and titles are represented correctly. Note that if any authors are added or removed after this point then all authors will be requested to provide approval documentation that could potentially delay the production of your paper.

Our response: We receive the comments with many thanks. The author list is complete and accurate now.

Comment 4: Article structure

We can accommodate up to 10 display items (Figures or Tables) in the main article. Each Figure and Table must fit easily within an A4 page (210 x 297 mm). Please ensure that the number and size of your Figures and Tables fulfil these requirements to avoid any delay in the acceptance of your article.

To comply with this format and optimise the presentation of data in your Article, we suggest the following changes to the display items in your paper: Please replace a.u. with Arb.Units.

Please ensure your main manuscript file includes the following sections, in this order:

Title

Author list

Affiliations

Abstract

Introduction

Results

Discussion (optional)

Results and Discussion (optional)

Methods (including Data Availability, Code Availability and Statistics subsections where relevant)

References

Acknowledgements

Author Contributions Statement

Competing Interests Statement

Tables

Figure Legends/Captions (for main text figures)

We do not edit Supplementary Information files; they will be uploaded with the published article as they are submitted with the final version of your manuscript. Any tracked changes should be removed from the file and the file should be provided as a PDF file. Supplementary Figures do not need to be provided separately.

Our response: We wish to thank the editor for these useful comments. First, five figures involved in the main manuscript meet the requirements. Second, we have changed the symbols “a.u.” in Figure 3a, 3c and Figure 5e to “Arb. Units”. Third, the manuscript is organized in the order of the following sections: Title, Author list, Affiliations, Abstract, Introduction, Results, Discussion, Methods, References, Acknowledgements, Author Contributions, and Competing Interests. Last but not the least, the symbols “a.u.” in the figures of Supplementary Information have been corrected. And the Supplementary Information file is uploaded in the PDF format.

Comment 5: Main text

Please remove acronyms and abbreviations from the abstract.

Please rearrange the Introduction so that all discussion of previous work appears first. The final paragraph should contain only a concise summary of the current work, in the present tense, and begin with a phrase like “In this work” or “Here, we show”.

Please do not use italics, bold font, underlining or speech marks unless required for technical terms (in both the main text and the display items).

Please make sure that mathematical terms throughout your manuscript and Supplementary Information (including in figures, figure axes, and legends) conform strictly to the following guidelines. Equations must be supplied in editable format, and not as images. Scalar variables (e.g. x , V , χ) must be typeset in italic, whereas multi-letter variables and functions (e.g. \log) must be formatted in roman. Vectors (such as the wavevector k or the magnetic field vector B) must be typeset in bold without italics.

Our response: We really appreciate the editor’s suggestions. First, the abbreviation “UCNCs” in

the abstract is replaced by “nanocrystals” (line 22, page 1). Second, according to the suggestion, we have modified the statement in line 63-70, page 2.

“In this work, we demonstrate that a UCNCs-doped size-mismatched photonic molecules (PMs) structure, evoking constructive Vernier effect in the resonance between the gain microcavity and loss one upon asymmetric pumping, gives rise to excellent unidirectional single-mode lasing. By harnessing Yb³⁺-Nd³⁺ co-doped UCNCs as the gain medium, dynamic lasing switch with an extremely large wavelength shift up to 300 nm is realized by external excitation modulation. This PMs laser in submillimeter scale was made by the CMOS-compatible photolithography technique, providing a smart and robust design to construct a mass-manufacturable compact lasing switch for environmental and photonic applications.”

Third, the formatting (i.e., italics, bold font, underlying and speech marks) are removed from the main manuscript and supplementary information. Last, the mathematical terms and the equations are modified according to the suggestions (see line 84, page 3; and line 248, page 9).

Comment 6: Figures and Tables

Please see the guidelines linked below for detailed instructions about how your figures should be prepared. Following these instructions will reduce the chances of delays should we need to request replacement artwork from you at a later stage. <https://www.nature.com/documents/NRJs-guide-to-preparing-final-artwork.pdf>

Please make sure that the terms ‘atomic units (a. u.)’ or ‘arbitrary units (arb. units)’ are appropriately used.

Any abbreviations, symbols or colours present in your figures must be defined in the associated legends.

In each Figure and Supplementary Figure where error bars are used, they must be defined.

Our response: We thank the editor for the very careful review and the valuable suggestions for revision. All the Figures in the main text and supplementary information are well prepared according to the guidelines. The symbols “a.u.” in Figure 3a, 3c and Figure 5e of the main text and supplementary figures are replaced by “Arb. Units”. And then, for the abbreviations, symbols, colours, and error bars, we have added the following statement in the main text:

“error bars in Figure 5c stand for the s.d.’s from four sets of measurement in the short-listed PMs device (line 339, page 13).”

“The highlighted regions in Figure (d,e) are intended to guide the eyes (line 341, page 13).”

“The distinct layers are highlighted by different colours (line 99, page 4).”

Comment 7: Data and Code

Nature journals strongly support public availability of data and code. Please deposit the data and code used in your paper into a public data repository, or alternatively, present the data as Supplementary Information. If data can only be shared on request, please explain why in your Data Availability Statement, and also in the correspondence with your editor.

Please note that for some data types, deposition in a public repository is mandatory. Any restrictions on sharing of these data types must be clearly indicated in the statement and discussed with the editor. More information on our data deposition policies and available repositories can be found here:

<https://www.nature.com/nature-research/editorial-policies/reporting-standards#availability-of-data>

All published manuscripts reporting original research in Nature Portfolio journals must include a data availability statement, as a separate section before the References and under the heading 'Data Availability'.

The data availability statement must make the conditions of access to the “minimum dataset” that are necessary to interpret, verify and extend the research in the article, transparent to readers.

This minimum dataset may be provided through deposition in public community/discipline-specific repositories, custom proprietary repositories or general repositories like Figshare, Zenodo and Dryad. Providing large datasets in supplementary information is strongly discouraged and the preferred approach is to make data available in repositories. Scientific Data, a Nature Portfolio journal, maintains a list of approved and recommended data repositories to support researchers seeking suitable repositories for their data (<https://www.nature.com/sdata/policies/repositories>).

The Data Availability Statement should also reference any source data published alongside the paper.

If DOIs are provided, we also strongly encourage including these in the Reference list (authors, title, publisher (repository name), identifier, year).

For clinical datasets or third party data, please ensure that the statement adheres to our policy (<https://www.nature.com/nature-research/editorial-policies/reporting-standards#availability-of-data>)

Please use the following template to provide all the information stated above:

The XX data generated in this study have been deposited in the YY database under accession code ZZ [add hyperlink here]. The XX data are available under restricted access for {insert reason}, access can be obtained by {explain how}. The raw XX data are protected and are not available due to data privacy laws. The processed XX data are available at YY. The XX data generated in this study are provided in the Supplementary Information/Source Data file. The XX data used in this study are available in the YY database under accession code ZZ [Add hyperlink here].

Specific advice on your Data Availability Statement: Please consider making this data available in a publicly accessible repository, or explicitly explain why the data can only be made available from the authors on request.

Our response: We thank the editor for these valuable comments. According to the suggestions, we have modified the statement in line 377, page 14 in the main manuscript.

“Data Availability: The supporting data generated in this study is publicly available in the online version of the paper. Reprints and permissions information is available online at [www.nature.com/reprints](http://www.nature.com/reprints). Correspondence and request for materials should be addressed to L.J. or X.C.”

Comment 8: Methods

Sufficient details of the experiments must be provided in the Methods section such that they could be reproduced without reference to published papers. Use of the term "as described previously" is not encouraged.

Centrifugation speeds must be reported in x g.

Please rename the Methods section as 'Methods'.

Our response: We thank the editor for the careful review and important suggestions. Accordingly, we have modified the statement in line 364, page 14 in the main manuscript.

“The multi-shell $\text{NaGdF}_4:\text{Yb}/\text{Nd}@\text{NaGdF}_4:\text{Yb}/\text{Ho}@\text{NaYF}_4:\text{Ca}@\text{NaYbF}_4:\text{Tm}@\text{NaYF}_4:\text{Ca}$ nanocrystals were synthesized according to the method in ref. 2. Detailed fabrication and characterization are provided in the Supplementary Information. ”

The centrifugation speed in section-1 of the supplementary information is changed from “6000 rpm” to “5760 g”. Beside, the “Materials and Methods” is replaced by “Methods” (see line 362, page 14) in the main text.

Comment 9: References

Supplementary References should appear at the end of the Supplementary Information file, and must be self-contained and numbered from 1. References mentioned in both the main text and the Supplementary Information should be part of both reference lists so that the Supplementary Information does not refer to the reference list in the main paper and vice versa.

Our response: We appreciate the editor for this valuable comment. According to the suggestion, supplementary references are deleted.

Comment 10: End matter

Nature Portfolio defines Competing Interest (CI) as financial and non-financial interests (including but not limited to funding, employment, stocks, shares, patents, personal or professional relationships with individuals or institutions, and unpaid membership advocacy) that could be perceived to directly undermine the objectivity, integrity, and value of a publication, or could be seen as having an influence on the judgments and actions of authors with regard to objective data presentation, analysis, and interpretation.

Please thoroughly review our policy on Competing Interests and include a detailed statement both in your final manuscript file and in our manuscript tracking system. Please ensure the statements are identical in both. Be specific about how each point stated relates to the research and list applicable author initials, and/or patent numbers.

If there are no competing interests, a negative statement must be included.

<https://www.nature.com/nature-research/editorial-policies/competing-interests>

Please confirm that all relevant funding awarded to each author is described in the Acknowledgements section. List each grant number, followed by the initials of the author who

received it.

Our response: We thank the editor for this careful review and important comments. The authors declare no competing interests. As for the relevant funding in the Acknowledgement section, Dr. L.J. is the person who receives the following grants: no.2018A030310034, 61805058, JCYJ20180306171700036, JCYJ20190806143813064, Shenzhen Scientific Research Foundation for the introduction of talent. Dr X.C. is in charge of the following grants: no.51802198, JCYJ20180507183532343, JCYJ20180507184613841, GXWD20201230155427003-20200821203750001, Shenzhen Scientific Research Foundation for the introduction of talent. Dr. Q.S. obtains the following grants: no. 12025402, 11974092, JCYJ20200109112805990, JCYJ20200109113003946, Fundamental Research Funds for the Central Universities. Dr. S.X. receives the following grants: no. 61975041, 11934012, JCYJ20210324120402006, Shenzhen Scientific Research Foundation for the introduction of talent.

Comment 11: Preparing your manuscript files

Unless otherwise stated please limit individual file sizes to approximately 30MB. We strongly encourage the use of repositories for large datasets or source data due to size considerations.

Please supply the main manuscript file in either Microsoft Word or LaTeX format. If you have written your manuscript in LaTeX, please upload the .tex file.

The use or adaptation of previously published images is strongly discouraged. If this is unavoidable, please request the necessary rights documentation to re-use such material from the relevant copyright holders and return this to us when you submit your revised manuscript. Please check whether your manuscript or Supplementary Information contain third-party images, such as figures from the literature, stock photos, clip art or commercial satellite and map data.

For more information on what constitutes ownership by a third party, please contact our Editorial Assistant at naturecommunications@nature.com

Our response: We thank the editor for this careful review and valuable suggestions. The size of each uploaded file (involving Editorial Policy Checklist, a point-by-point response to the reviewer's and editor's comments, a complete of author checklist, the main article with tracked changes and with tracked changes accepted, separate figure files, inventory of supporting information, and a supplementary information) is less than 30 MB. Note that, the main manuscript file is uploaded in Microsoft Word format. No images in this manuscript are served by third parties.

Comment 12: Forms to complete

Editorial Policy Checklist

Please update and upload a final version of the Editorial Policy Checklist with your revised manuscript files. A blank Editorial Policy Checklist can be found via the link below. Note that this form is a dynamic 'smart pdf' and must be downloaded and completed in Adobe Reader.

Please update your current checklist or download from:

<https://www.nature.com/documents/nr-editorial-policy-checklist.zip>

Our response: We receive the comment with many thanks. The revised Editorial Checklist is completed and uploaded.